# Winter Arctic sea ice thickness from ICESat-2: upgrades to freeboard and snow loading estimates and an assessment of the first three winters of data collection

Alek A. Petty[1, 2], Nicole Keeney[1, 2], Alex Cabaj[3], Paul Kushner[3], Marco Bagnardi[1, 4]

[1] Cryospheric Sciences Laboratory, NASA Goddard Space Flight Center, Greenbelt, MD, USA.
[2] Earth System Science Interdisciplinary Center, University of Maryland, College Park, MD, USA.
[3] University of Toronto, Toronto, Canada.
[4] ADNET Systems Inc., Bethesda, MD, USA.

*Correspondence to*: Alek A. Petty (alek.a.petty@nasa.gov)

**Abstract**

NASA's ICESat-2 mission has provided near-continuous, high-resolution estimates of sea ice freeboard across both hemispheres since data collection started in October 2018. This study provides an impact assessment of upgrades to both the ICESat-2 freeboard data (ATL10) and NASA Eulerian Snow On Sea Ice Model (NESOSIM) snow loading on estimates of winter Arctic sea ice thickness. Misclassified leads were removed from the freeboard algorithm in the third release (rel003) of ATL10, which increased freeboards in January and April 2019, and increased the fraction of low freeboards in November 2018, compared to rel002 data. The thickness increases due to increased freeboards in ATL10 improved comparisons of Inner Arctic Ocean sea ice thickness with thickness estimates from ESA's CryoSat-2 and show good agreement on basin-average thickness changes over multiple winter seasons, depending on the specific product analysed. The upgrade from NESOSIM v1.0 to v1.1 results in only small changes in snow depth and density which have a less significant impact on thickness compared to the rel002 to rel003 ATL10 freeboard changes. The updated monthly gridded thickness data are validated against ice draft measurements obtained by upward looking sonar moorings deployed the Beaufort Sea, showing strong agreement ($r^2$ of 0.87, differences of 11 +/- 20 cm). The seasonal cycle in winter monthly mean Arctic sea ice thickness shows good agreement with various CryoSat-2 products (and a merged ICESat-2/CryoSat-2 product) and PIOMAS. Finally, changes in Arctic sea ice conditions over the past three winter seasons of data collection (November 2018 - April 2021) are presented and discussed including a 50 cm decline in multi-year ice thickness and negligible interannual differences in first-year ice. Interannual changes in snow depth provide a notable impact on the thickness retrievals on regional and seasonal scales. The gridded data analysis is provided online in a Jupyter Book format to increase transparency and user engagement with our upgraded monthly gridded winter Arctic sea ice thickness dataset.

## 1 Introduction

Sea ice provides an important role in our climate system by modulating energy exchange between the atmosphere and ocean and controlling the biogeochemical balance of the polar oceans (Aagaard and Carmack, 1989; Serreze et al., 2007; Post et

al., 2013). The areal coverage and thickness of sea ice also strongly influences human and marine activities, especially in the Arctic (Eicken, 2013; Cooley et al., 2020). Reliable basin-scale observations of polar sea ice are thus urgently needed to better understand recent changes and constrain future projections of sea ice conditions.

Passive microwave sensors now provide an over 40 year record of sea ice area and extent changes at daily time-scales (Parkinson and DiGirolamo, 2021) – data which can be used to explore the regional, seasonal and interannual variability of sea ice, including the ~50% decline in September Arctic sea ice area in recent decades. However, observations of sea ice thickness are generally more limited as (i) polar-focussed satellite altimetry missions only started in earnest with the launch of NASA's Ice, Cloud, and Land Elevation Satellite (operated between 2003 and 2009, Schutz et al., 2005) and ESA's CryoSat-2 (in operation since 2010, (Wingham et al., 2006)), (ii) derivations of sea ice freeboard from altimetric height observations are challenging due, for example, to the need to accurately distinguish leads/cracks from ice to derive a local reference sea surface, and (iii) converting altimetric estimates of sea ice freeboard to thickness requires various additional data assumptions, including the state of the overlying snow loading, which is poorly constrained, especially over the Southern Ocean.

NASA's Ice, Cloud, and Land Elevation Satellite-2 (ICESat-2) was launched in 2018 and included a key objective to *estimate sea-ice thickness to examine ice/ocean/atmosphere exchanges of energy, mass and moisture* (Markus et al., 2017). The Advanced Topographic Laser Altimeter System (ATLAS) onboard ICESat-2 measures surface elevation at high resolution (individual laser footprints of ~11 m, Magruder et al., 2020) and high precision (< 2 cm over sea ice flat surfaces, Kwok et al., 2019a), with dense along-track sampling (70 cm along-track from the 10 kHz pulse repetition rate, Neumann et al., 2019). ATLAS was designed in part to obtain accurate and routine estimates of total (ice plus snow) freeboard, the vertical extension of sea ice and its overlying snow cover above local sea level, across the polar oceans (Markus et al., 2017). Sea ice freeboard can typically range from millimeters to tens of centimeters depending on the region or season profiled. ICESat-2 benefits from extensive polar coverage (profiling up to 88 degrees N/S, monthly sub-cycle) and has collected year-round data with minimal downtime since production started in October 2018. ICESat-2 sea ice height and freeboard data are provided in the official ATL07 (Kwok et al., 2021a) and ATL10 (Kwok et al., 2021b) products respectively. The first winter season of ICESat-2 Arctic Ocean freeboards (ATL10) was presented in Kwok et al., (2019b), highlighting the regional and seasonal freeboard distributions obtained by ICESat-2.

Validation of the ATL07 and ATL10 products is on-going. ATL07 sea ice heights showed very strong agreement (0 cm mean differences, correlation coefficients of 0.97 to 0.98) with coincident airborne data collected by NASA's Operation IceBridge north of Greenland and the Canadian Archipelago in spring 2019 (Kwok et al., 2019a). The freeboard agreement was more modest (mean differences of 0 to 4 cm), although the comparisons were hindered by the lack of available leads to reliably determine a local sea surface in either product. Additional analysis of the ATL07/10 surface classification scheme using imagery collected by the Copernicus Sentinel-2 mission, provided evidence of high skill in lead classification during cloud-free/day-time conditions  (Petty et al., 2021), a key part of the freeboard determination procedure. However, both the spring 2019 OIB Arctic campaign comparisons (Kwok et al., 2021d) and Sentinel-2 imagery assessments (Petty et al., 2021)

highlighted errors in the 'dark lead' classification in ATL07/10. Briefly, it was hypothesized that low/optically thin clouds in these regions attenuate the photon rate around these segments due to increased atmospheric scattering, tricking the empirical threshold-based classification algorithm into characterizing height segments over sea ice as dark leads. High photon rate specular leads are now the only lead types used to derive sea surface and thus freeboard in the Release 003 and subsequent sea ice products (Release 005 at the time of writing) while a possible filter for the dark-lead segments is being developed and tested. The impact of this change was an increase in freeboard in ATL10 of 0 to 3 cm depending on the season/region analyzed, as well as a decrease in coverage due to the reduction in sea surface tie-points (Kwok et al., 2021d).

Measurements of freeboard are typically obtained for the goal of estimating sea ice thickness, see schematic in Figure 1. This is conventionally achieved by combining freeboard measurements with ancillary estimates of snow loading (snow depth and density), sea ice density and an assumption of hydrostatic equilibrium (e.g. Giles et al., 2007; Kwok and Cunningham, 2008; Laxon et al., 2013; Kwok, 2018). Sea ice thickness was estimated from Release 002 ATL10 freeboards using external snow loading estimates from the NASA Eulerian Snow on Sea Ice Model (NESOSIM) v1.0 and modified versions of the Warren et al. (1999) snow climatology (Petty et al., 2020, P2020). The Feb/March 2019 ICESat-2 thicknesses were ~10 cm thinner than Feb/Mar 2008 ICESat thickness estimates, alluding to a possible decline in end-of-winter Arctic sea ice thickness over this 11-year period. However, the P2020 thickness estimates were also thinner than those produced using radar freeboard measurements from ESA's CryoSat-2 using the same input assumptions (tens of cm biases depending on the month and product analysed). Significant biases still exist in satellite-derived estimates of sea ice thickness, even those based on the same satellite sensor, e.g. radar altimetry data from ESA's CryoSat-2 mission (Sallila et al., 2019; Petty et al., 2020) which have limited their utility to-date, e.g. for constraining or calibrating polar climate projections (e.g. SIMIP Community, 2020).

The thickness results presented in P2020 used NESOSIM v1.0 snow loading forced by the European Centre for Medium-Range Weather Forecasts (ECMWF) ERA-Interim (ERA-I) snowfall (Dee et al., 2011). However, ERA-I production ended in August 2019 and was superseded by ERA5 (Hersbach et al., 2020). While ERA5 total precipitation is similar to ERA-I over the Arctic Ocean (Wang et al., 2019; Barrett et al., 2020), ERA5 produces relatively more snowfall and thus less rainfall compared to ERA-I, especially in the Atlantic sector of the Arctic (Wang et al., 2019). Additional developments and calibration of NESOSIM have been carried out to upgrade NESOSIM (v1.0 to v1.1) and extend the derived ice thickness product beyond the first winter season (2018/2019) presented in P2020 which we present here.

The significant changes in ATL10 freeboards and the availability of updated NESOSIM snow loading warrants an updated winter Arctic sea ice thickness assessment. ATL10 and NESOSIM v1.1 output are now (at the time of submission) also available from fall 2018 through to spring 2021, providing three winter seasons of data to assess. The main objectives of this paper are to: (i) highlight upgrades to the ICESat-2 ATL10 freeboard product and NESOSIM v1.1 snow loading and assess their impact on winter Arctic sea ice thickness; (ii) carry out updated comparisons against CryoSat-2 derived thickness estimates and newly released ice draft measurements; and (iii) assess monthly gridded thickness data from the past three winter seasons across the entire Arctic Ocean.

## 2 Data and Methods

### 2.1 ICESat-2 ATL10 freeboards

We use the ICESat-2 ATL10 freeboard product (currently at Release 005, rel005), which is disseminated through the National Snow and Ice Data Center (NSIDC) (Kwok et al., 2021a). ATL10 is the end result of a series of algorithms that convert the primary geolocated photon product (ATL03, Neumann et al., 2019), to sea ice height and type (ATL07, Kwok et al., 2021a), and then sea surface height and freeboard (ATL10, Kwok et al., 2021b). Briefly, the ATL07 algorithm subtracts a mean sea surface and time-varying ocean tide and inverted barometer corrections from ATL03, then aggregates and windows 150 photons around this corrected surface along each beam independently. ATL07 then extracts a best-guess Gaussian height distribution convolved with the expected system response to the photon height histogram to determine a single 'segment' height and various metrics summarizing the goodness of fit and radiometry (e.g., photon rate) of each segment. This photon aggregation results in data with variable segment lengths of, on average, ~15 m for the strong beams and ~60 m for the weak beams (Kwok et al., 2019b). The spatial resolution of the individual segments can be estimated by adding the individual laser footprint size of ~11 m (Magruder et al., 2020) to the segment length, i.e., a mean of ~25 m for the strong beams and 70 m for the weak beams. An empirically based decision-tree algorithm is used to discriminate the height segments as either sea ice or sea surface/lead (Kwok et al., 2016). More details of the surface classification scheme are available in Kwok et al., (2021d) and Petty et al., (2021), while the complete processing methodology is available in the Algorithm Theoretical Basis Document (ATBD) for sea ice products (Kwok et al., 2021c).

ATL10 converts adjacent sea surface segments into lead groups to reduce noise in the lead height estimate and then averages these into 10 km along-track sea surface reference height estimates along each beam. Freeboard is calculated as the difference between the individual ice height segments and the local sea surface height, independently for each beam. Negative freeboards are set to zero. The laser returns are expected to track the snow-covered ice surface, so ATL10 is expected to provide a measure of 'total' (ice plus snow) freeboard. The ICESat-2 beams are arranged in 'strong' and 'weak' beam pairs with each beam pair separated by ~3.3 kilometers in the across-track direction and the strong/weak beams separated by ~90 m across-track and ~2.5 km along-track. The weak beams are around 4 times lower energy (lower photon rate) than the strong beams. In this study we utilize only the strong beams to ensure the highest possible data quality.

### 2.1.1. ATL10 upgrades

The ICESat-2 sea ice products are continuously being updated as new assessments on the data are undertaken. All ICESat-2 products currently follow the same nominal release schedule (~6-12 months), so release updates are not necessarily based on the significance of the changes or improvements made to the given product. All new release data are processed and released from the start of the mission (October 14[th], 2018) onwards, until the production of a new release begins. The sea ice thickness results presented in P2020 utilized rel002 ATL10 data, and differences with thickness estimates produced using rel001 ATL10 were noted to be negligible. As discussed earlier, in rel003 ATL10 and subsequent releases, dark leads have

been removed as possible sea surface height segments, since false positive classifications were found in the presence of clouds, resulting in an increase in basin averaged freeboards of up to 3 cm and some loss in coverage, especially within the more consolidated central Arctic ice pack (Kwok et al., 2021d). This is arguably the biggest change in the ICESat-2 sea ice products to-date. The rel003 ATL10 data also included a relaxing of the height/freeboard quality flag (from 3 to 4), which means height segments with a poorer fit, generally segments from ridges with a more variable and complex height profile, are now included to increase retrieval counts over ridged ice regimes.

In rel004, most of the updates involved changes related to the treatment of the solid earth tides - a transition of ATL07 into a tide-free system to be consistent with ATL03. This caused a significant change in the magnitude of the heights reported in ATL07 and ATL10, but as freeboard is a relative measurement, this was not expected to impact the reported freeboards. In rel005 ATL10, the only changes relevant to freeboard determination include improved calculation of the 10 km reference surface location to the centre of each section (effectively a bug fix). The rel005 data now also includes data from previously held granules where known satellite calibration scans were occurring somewhere along the granule. New automated pointing angle and calibration scan filters were introduced in rel005 to ensure only data within each granule experiencing degraded performance are filtered out, instead of withholding entire data granules. Most other developments in rel003 to rel005 ATL10 can be categorized as minor bug fixes and are listed in the ATBD change log, made available since rel004 (Kwok et al., 2021c). New releases of ATL07 and ATL10 also reflect upgrades to the underlying ATL03 processing, such as improvements in geolocation.

In Figure 2 we show the coverage change from rel002 to rel005 by counting the number of 10 km sea surface reference tie points available across the 4 releases from all data collected by the strong beams between November 2018 and April 2019. The figure highlights the strong decline in coverage between rel002 and rel003. The rel003 to rel005 coverage differences are sporadic and linked mainly to the inclusion of calibration scan data granules. An expanded figure showing all release differences is given in the Supplementary Materials (Figure S1). Calibration scans occur mainly over lower latitudes but can occasionally extend over the Arctic ice pack – data during these scans are generally considered degraded i.e., heights with sub-nominal geolocation quality. Automated calibration scan filtering was introduced in rel005 to exclude these data more reliably and ensure only the highest quality height returns are utilized. In Figure S2 we provide a beam coverage assessment over the same time period using rel005 data only, highlighting the consistently higher coverage provided by the strong beams compared to the weak beams across this first winter of data collection). The middle beam pair is notable for the higher reference counts compared to other strong and weak beams .

## 2.2 NESOSIM

We use snow depth and density estimates from the NASA Eulerian Snow On Sea Ice Model (NESOSIM) (Petty et al., 2018a, P2018) which is publicly available on GitHub (https://github.com/akpetty/NESOSIM). NESOSIM was developed primarily in preparation for the launch of ICESat-2, to enable timely production of snow depth and density estimates for sea-ice thickness retrievals using a simple snow accumulation model framework. NESOSIM includes two vertical snow layers

and several simple parameterizations (accumulation, wind packing, advection–divergence, blowing snow loss) to represent the expected primary sources and sinks of snow on Arctic sea ice during the accumulation season. Summer melt processes are currently neglected, so the model is typically run between September and the end of April. NESOSIM v1.0 was first presented in P2018 and the output using this v1.0 framework was used in P2020 to produce snow loading needed to convert ATL10 freeboards (rel002) to sea ice thickness from October 2018 to April 2019. The NESOSIM v1.0 output used in P2020 was forced with snowfall, winds and near-surface air temperature (to scale the initial snow conditions) from ERA-I (Dee et al., 2011), sea ice concentrations from the NASA Climate Data Record (CDR) version 3 (Meier et al., 2017), and ice drifts from the European Organization for the Exploitation of Meteorological Satellites (EUMETSAT) Ocean and Sea Ice Satellite Application Facility (OSI SAF) (Lavergne et al., 2010) which were all regridded to a 100 km x 100 km Arctic Ocean domain using bilinear interpolation.

### 2.2.1. NESOSIM upgrades

Here we describe recent upgrades made to NESOSIM which has been tagged as a new version 1.1 (v1.1) code release (https://github.com/akpetty/NESOSIM/releases/tag/v1.1, archived at https://doi.org/10.5281/zenodo.4448356). Key updates in NESOSIM v1.1 include: CloudSat snowfall scaling (Cabaj et al., 2020, and described more below), a new blowing snow atmosphere loss term, an extended Arctic domain to cover the full extent of the Arctic peripheral seas, an improved smoothing filter to reduce noise in the dynamic snow budget terms, an upgrade to Python 3, and various minor bug fixes. Much of the NESOSIM v1.1 development was motivated by the need to recalibrate NESOSIM using ERA5 forcings (Hersbach et al., 2020), now that ERA5 has succeeded ERA-I following the end of ERA-I data production in August 2019 and given reports of increased ERA5 snowfall compared to ERA-Interim (Wang et al., 2019; Cabaj et al., 2020). ERA5 is thought to offer improvements over ERA-I related to improved cloud representation, an updated assimilation scheme and higher spatial resolution (Hersbach et al., 2020). Regardless, whether ERA5 exhibits a high snowfall bias over the Arctic or ERA-I a low bias is still uncertain and likely regionally dependant. Cabaj et al., (2020) used snowfall estimates from CloudSat to calibrate several reanalyses snowfall estimates, including ERA5, within the NESOSIM framework – reducing the spread in snowfall from the chosen reanalyses, although not significantly changing the magnitude of the ERA5 snowfall in the North Atlantic region, where winter snowfall rates are highest overall. On average, ERA5 reports more snowfall over the Arctic basin than what is observed by CloudSat measurements, so the scaling tends to slightly decrease the overall magnitude of the snowfall and the resulting snow depth in NESOSIM (Cabaj et al. 2020). The CloudSat-reanalysis scaling coefficients are now included in the NESOSIM v1.1 code repository.

The other significant code development was the introduction of a new blowing snow loss term. The simple parameterization of blowing snow lost to leads/open water introduced in NESOSIM v1.0 is given as

$$\Delta h_s{}^{bs\_ow} = -(1 - A)\beta\, T_d U\, h_s \text{ for } U > \omega \quad [1]$$

where $A$ is ice concentration, $h_s$ is the snow depth in the top 'new' snow layer, $U$ is the wind speed, $T_d$ is the number of seconds in the daily time-step, $\omega$ is the wind action threshold, $\beta$ is the blowing snow loss coefficient. This parameterization has been challenged due to uncertainties around how much snow might be lost to open water under windy conditions, rather than sublimated, i.e. lost to the atmosphere, or transported either within or to adjacent grid-cells (Liston et al., 2020).

Motivated by this, we introduced an additional blowing snow atmosphere loss term, which is a similar function of wind speed and snow in the top 'new' snow layer to the loss-to-open water term, but not also a function of sea ice concentration:

$$\Delta h_s^{bs\_a} = -\gamma\, T_d U\, h_s \text{ for } U > \omega \text{ [2]}$$

where $\gamma$ is a new blowing snow atmosphere loss coefficient. This parameterization, which provides a simple mechanism for increasing snow loss under given atmospheric conditions independent of sea ice conditions, requires calibration of an additional free parameter, $\gamma$, which we discuss below. As discussed in the original NESOSIM study (Petty et al., 2018), these snow loss terms are crude representations of complex physical processes that we introduce primarily to remove snow and improve correspondence with the limited observations we have for calibration purposes.

Additionally, NESOSIM v1.1 was forced with daily sea ice concentrations from the NASA Climate Data Record (CDR) version 3 (Meier et al., 2017), daily ice drifts from both the NSIDC Polar Pathfinder version 4 dataset (Tschudi et al., 2019) from 1980 to April 2019 and daily drifts from the European Organization for the Exploitation of Meteorological Satellites (EUMETSAT) Ocean and Sea Ice Satellite Application Facility (OSI SAF) global low resolution ice drift dataset (Lavergne et al., 2010) from September 2019 to April 2021 due to contrasting data availability. As noted in P2018, the

impact on snow depth from ice drift forcing is generally second order to snowfall, although this can have first-order impacts at more regional scales. All forcings were regridded to our updated 100 km x 100 km North Polar Stereographic (EPSG: 3413, https://epsg.io/3413) Arctic Ocean model domain.

We recalibrated NESOSIM v1.1 considering the new forcings and model changes described above, by targeting estimates of spring Arctic snow depths derived from Snow Radar data collected during NASA's Operation IceBridge as used

in P2018: the snow radar layer detection (SRLD) product (Koenig et al., 2016), the NASA Goddard Space Flight Center (GSFC) empirical threshold based product (Kurtz et al., 2013) and the Jet Propulsion Laboratory (JPL) product (Kwok et al., 2017). Our approach differs from the calibration approach used in P2018, which calibrated NESOSIM v1.0 against Soviet Station drifting station data collected in the 1980s (Warren et al., 1999) then assessed these results against OIB-derived snow depths. Here we choose instead to recalibrate NESOSIM v.1.1 against the spring OIB snow depth data from 2010 to 2015 to

provide a more reliable snow depth representation focussed on our contemporary period of interest. We retain, however, the density values for the new 'top' and old 'bottom' layer snow (Table 1) which were derived from the Soviet Station calibration effort.

As noted in P2018 and presented in (Kwok et al., 2017), there is a large spread between the available OIB snow depth products due to various challenges in interpreting Snow Radar data. To account for this large inter-product uncertainty,

we use the median gridded OIB spring snow depth from these three datasets. Specifically, we take all raw (~7 m along-track resolution) snow depth measurements from the three snow depth retrieval algorithms for a given day, grid them to the 100 km x 100 km NESOSIM v1.1 Arctic Ocean model domain using a simple binning procedure (average of all snow depths in the given grid-cell in each day), then take the median snow depth value at each daily grid-cell across the three OIB products. Quick-Look (QL) snow depths are available for the more recent years (2012 to 2019), using the GSFC waveform fitting approach (https://nsidc.org/data/NSIDC-0708/versions/1). However, it was noted in Kwok et al., (2017) that these estimates tend to exhibit a low bias compared to the other OIB products. A low bias in the GSFC QL product was also shown based on in-situ measurements collected in March 2014 (King et al., 2015). These biases were confirmed in our own analysis comparing our median OIB snow depths with the GSFC QL product (2013-2015), showing mean biases of ~6 cm (QL thinner than the gridded median data, see Supplementary Figure S3), motivating us to exclude these from our model calibration efforts here.

We heuristically calibrated NESOSIM v1.1 using the daily OIB gridded median snow depths with the aim of removing the mean bias relative to OIB when using the default NESOSIM v1.0 parameter settings (Figure 3a). Current work is exploring more automated calibration approaches (Cabaj et al., 2021), but here we were able to find a solution that reduced the mean bias to 0 cm by halving the blowing snow open water coefficient, extending the model initialization date to September 1 instead of August 15 and tuning the new atmosphere snow loss coefficient, $\gamma$ (Eq. 2) as shown in Figure 3b. In the absence of contemporary early-season ground-truth data, we view the initial conditions (either their distribution or the representative start date) as another tuning parameter, constrained mainly by limited evidence in the literature. For example, the Warren et al., (1999) climatology (W99) shows a mean snow depth of 3 cm in August including depths of up to 8 cm near the Greenland/Canadian Arctic coastline based on the quadratic fit to observations. However, output from SnowModel-LG presented in Stroeve et al., (2020) shows zero snow depths in August in the earlier (1985/1986) and later (2015/2016) time periods of that time-series. As NESOSIM includes no snow melt terms, we prefer instead to initialize later in the year (Sep 1st) and prescribe an expected end of August mean snow depth based on the temperature scaled W99 August climatology shown in Petty et al., (2018, see Figure 2). NESOSIM v1.1 was run from 1980 to 2021 and is expected to be updated in future years to enable continued thickness processing from ICESat-2.

The output from this v1.1 model framework from 1980-2021 has been archived on Zenodo (https://doi.org/10.5281/zenodo.5164314). The NESOSIM v1.0 output from P2018 was originally released from 2000 to 2015 only but was extended for the 2018/2019 winter to produce snow depths used in the initial P2020 ICESat-2 sea ice thickness processing.

Figure 4 shows a time-series comparison of the October and April mean snow depths from NESOSIM v1.0 and v1.1 within an Inner Arctic Ocean domain (Central Arctic, Beaufort Sea, Chukchi Sea, Laptev Sea, Kara Sea, region map provided in Figure 5). NESOSIM v1.1 shows good agreement with v1.0 in terms of the October and April mean snow depth and patterns of interannual variability. Differences between the two releases are <5 cm and often near zero. The longer record of NESOSIM v1.1 output is strongly suggestive of a long-term decline in snow depth and near-record low snow

depths in recent years, i.e., the ICESat-2 period 2018-2021, however a snow trend analysis is beyond the scope of this study.

P2020 noted strong differences between NESOSIM v1.0 and mW99 (mW99, snow depths halved over first-year ice, see Supplementary Figures S2 to S4 in P2020). Generally, NESOSIM v1.0 snow depths are similar over the thicker multiyear ice, but mW99 is thinner later in the year, due primarily to the thinner snow over first-year ice. Figure 4 shows the mean Inner Arctic Ocean snow depth from mW99 (snow depths halved using observed OSI SAF ice type data for the given month/year), showing similar values to NESOSIM v1.1 (and v1.0) in October but thinner mW99 snow in April. The

interannual variability in mW99 is a result of variability in the monthly ice type mask (defining which grid-cells are modified). In October the mW99 depth variability appears similar to NESOSIM, however in April this variability appears weaker than NESOSIM, although this comparison is limited by the availability of OSI SAF ice type data needed to derive the mW99 estimates. Spatial difference maps between NESOSIM v1.1 and mW99 for the 2018-2021 ICESat-2 time period presented here are shown in the Supplemental Figure S4 highlighting the slightly thinner NESOSIM v1.1 snow depths in

October compared to mW99 but generally thicker snow in April, especially in the Kara Sea – broadly in-line with the NESOSIM v1.0 comparisons given in P2020. The difference between NESOSIM v1.0 and v1.1 snow density is minimal (as this was not the focus of the v1.1. upgrades) and is expected to have a negligible impact on our thickness results, so we opt against an additional density comparison here. To place the ICESat-2 period results in broader context, Figure 4 shows the monthly mean NESOSIM v1.1 snow depth distributions as violin plots, with the recent ICESat-2 years overlaid. In the initial

accumulation months of September/October, recent years show similar or deeper than average snow, while the middle/end-of-winter months, November to April, show clearly thinner than average snow in the recent ICESat-2 years. The 2019-2020 and 2020-2021 snow depths especially are at or near record low values across most of these months, with 2020-2021 April at the record low, while 2019-2020 and 2018-2019 are instead near to the mean. Capturing this interannual variability was the key motivating factor behind the development of NESOSIM and its use in the thickness processing which we discuss more

in the following sections.

## 2.3 ICESat-2 sea ice thickness data upgrades

We use the same approach as in P2020 to generate estimates of winter Arctic sea ice thickness and an associated uncertainty estimate. Briefly, thickness is calculated assuming hydrostatic equilibrium and input estimates of sea ice density, snow depth and snow density. The coarse resolution (~100 km) snow depth input estimate, primarily from NESOSIM, is redistributed to

the high-resolution (~30 m) ATL10 freeboards using a piecewise functional fit obtained from snow depth and freeboard data collected by NASA's Operation IceBridge mission (Kurtz et al., 2009; Petty et al., 2020). Uncertainties are calculated by propagating errors through the hydrostatic equilibrium equation with contributions from random errors (estimates based on previous studies) and systematic errors (estimates based on the spread in applied input assumptions). Small differences in our thickness processing to that presented in P2020 include a bilinear interpolation scheme instead of nearest neighbour to assign

NESOSIM data to the ATL10 freeboard segments. Nearest neighbour interpolation was originally used to reduce processing time but introduces unphysical step changes. We also fixed some minor bugs in the freeboard uncertainty calculation and

have incorporated the new NSIDC regional mask of the Arctic Ocean (Figure 5). As in P2020 we use daily estimates of ice type from the European Organization for the Exploitation of Meteorological Satellites (EUMETSAT) Ocean and Sea Ice Satellite Application Facility (OSI SAF, www.osi-saf.org) (Breivik et al., 2012) to classify each segment as either first-year ice (FYI) or multiyear ice (MYI). Ice type information is needed in-part to derive the modified Warren snow depth estimates (see Section 2.2.2. in P2020), so our approach is to assume all ice is MYI unless the OSI SAF product explicitly characterizes the segment as FYI. Thus, in September when OSI SAF does not provide any ice type estimate due to added uncertainties in the end-of-summer retrievals, we assume all our ATL10, and derived thickness data are MYI. The along-track thickness data, both raw segment-scale data and 10 km means, have been made available through the NSIDC (IS2SITDAT4 Version 1, https://nsidc.org/data/is2sitdat4/, Petty et al., 2022 ). We plan to update this dataset each year as new winter Arctic ATL10 data are generated and released.

In producing the monthly gridded dataset, we use all three strong beams to increase coverage and lower expected uncertainties, compared to the single strong beam used in P2020. The use of all three strong beams was also motivated by the reduction in data coverage in rel003 and onwards ATL10 data processing (described in Section 2.1.1 and noted in Figure 2). Our gridding approach is slightly different from the official gridded ICESat-2 freeboard product (ATL20, https://nsidc.org/data/ATL20) as we bin all data within a given month for each grid-cell, as opposed to producing daily gridded composites then monthly gridded composites from the daily gridded data. Our monthly gridded data includes ancillary data variables representative of the mean day of the month for each grid-cell calculated as the mean date of the input ATL10 data, and the number of ATL10 freeboard segments used in the monthly grid-cells to enable sampling bias assessments. The monthly gridded data also includes monthly NOAA/NSIDC Version 4 Climate Data Record (CDR) sea ice concentrations (Meier et al., 2021b), the new NSIDC regional mask of the Arctic Ocean and the OSI SAF ice type mask (sub-sampled by ICESat-2 then gridded monthly). The data are projected on to the NSIDC North Polar stereographic grid (EPSG: 3411, https://epsg.io/3411) and binned onto a 25 km x 25 km grid.

The initial version of our monthly gridded thickness dataset as described in P2020 was made available through the NSIDC (IS2SITMOGR4 Version 1, https://nsidc.org/data/IS2SITMOGR4). Our updated dataset presented in this study using rel005 ATL10 data, NESOSIM v1.1 and the updated NSIDC Arctic region mask and CDR sea ice concentrations have been made available as a new Version 2 (v2) release of the IS2SITMOGR4 dataset (Petty et al., 2022b). We expect to update this each year along with the along-track product (IS2SITDAT4) as new winter Arctic ATL10 data are made available. We also include in this IS2SITMOGR4 v2 dataset smoothed and interpolated variables of freeboard, snow depth and thickness in an initial attempt to fill in the pole hole and mitigate the spatial sampling biases. These preliminary variables are not used in the subsequent analysis presented here but their derivations are described and made available to interested users in the online Jupyter Book discussed below. We expect that future work will explore more sophisticated interpolation procedures and blending with other thickness datasets, which we discuss more in the summary section.

## 2.4 CryoSat-2 sea ice thickness estimates

In P2020, monthly gridded ICESat-2 thickness data were compared with thickness estimates generated from the European Space Agency (ESA) CryoSat-2 mission from four different groups: NASA's Goddard Space Flight Center (GSFC, Kurtz and Harbeck, 2017), the Jet Propulsion Laboratory (JPL, Kwok and Cunningham, 2015), the Center for Polar Observation and Modelling (CPOM, Laxon et al., 2013; Tilling et al., 2018) and the Alfred Wegener Institute (AWI, Hendricks and Ricker, 2016). Large differences were noted on these regional/monthly scales (ICESat-2 generally thinner), so

here we repeat the comparison using upgraded monthly gridded thickness data. As the different products make different assumptions regarding snow loading and sea ice, for these comparisons we use the same snow loading assumptions in the ICESat-2 thickness processing to generate direct thickness comparisons to simplify the interpretation of the differences observed (differences should be due to differences in the freeboard retrievals rather than differences in input assumptions). As in P2020, we re-grid the monthly gridded CS-2 estimates to the NSIDC 25 km x 25 km North Polar Stereographic grid

using a simple nearest neighbor interpolation scheme and compare these with our gridded ICESat-2 sea ice thickness estimates that have been produced using the same snow loading and ice density assumptions as the given CS-2 product, as summarized in Table 2 in P2020. Modified versions of the Warren snow depth climatology (mW99, Warren et al., 1999) were used by all four of these CryoSat-2 thickness products. Differences between mW99 and NESOSIM are discussed in Section 2.2.1. To be consistent with the results shown in P2020 we just use strong beam 1 to generate these data. This

comparison is not seen as a validation of either product, but a way of simply exploring regional/seasonal differences between these products due to the contrasting freeboard retrievals.

As we seek in this study to also track seasonal winter changes in sea ice thickness, we carry out an additional CryoSat-2 comparison of monthly mean Inner Arctic Ocean sea ice thickness between our gridded thickness estimate and various CryoSat-2 thickness estimates. The goal here is to simply highlight the correspondence (or lack thereof) in the winter

seasonal thickness cycles for the years of overlap. In this comparison we compare our best-guess ice thickness (e.g., ICESat-2 thickness using NESOSIM v1.1 snow loading) with the various CryoSat-2 products. We utilize the same CPOM and GSFC products as the earlier comparison, which have been updated through to 2021. We then use an upgraded AWI product that incorporates SMOS estimates of thin sea ice to improve both accuracy and coverage over thinner first-year ice (AWISMOS, Ricker et al., 2017). These data have been used together with our gridded ICESat-2 thickness data to explore recent sea ice

thickness/volume changes for the NOAA Arctic Report Card (Meier et al., 2021a). We also utilize a newly released all-season CryoSat-2 thickness product developed by University of Bristol (UBRIS, Landy et al., 2022) that utilizes snow loading from SnowModel-LG (Liston et al., 2020) and provides data through to summer 2020. Finally, we also show comparisons with thickness data generated from the combination of ICESat-2 and CryoSat-2 data, utilizing the assumption that ICESat-2 profiles total (ice plus snow) freeboard and CryoSat-2 profiles ice freeboard to then derive snow depth and

thickness concurrently from the combined altimetry record (KK, Kwok et al., 2020; Kacimi and Kwok, 2022).

## 2.5 PIOMAS sea ice thickness estimates

We additionally compare the ICESat-2 and CryoSat-2 monthly mean Inner Arctic Ocean sea ice thickness with those generated from the Pan-Arctic Ice-Ocean Modeling and Assimilation System (PIOMAS, v2.1; Zhang & Rothrock, 2003). PIOMAS is an ice-ocean model that generates estimates of sea ice thickness, constrained predominantly by the assimilation of sea ice concentration and sea surface temperature. PIOMAS data is commonly used in the sea ice community for assessments of Arctic sea ice thickness variability at regional and basin-scales (Tilling et al., 2015; Labe et al., 2018; Petty et al., 2018b; Schweiger et al., 2021; Moore et al., 2018). PIOMAS ice thicknesses estimates have been shown to exhibit differences on the order of tens of centimeters compared to satellite-derived estimates, although this depends strongly on the season and region analyzed (Schweiger et al., 2011; Zygmuntowska et al., 2014; Petty et al., 2018b).

## 2.6 Beaufort Gyre Exploration Projection (BGEP) upward looking sonar draft measurements

More recently, upward looking sonar (ULS) draft data from moorings deployed as part of the Beaufort Gyre Exploration Project (BGEP) covering the period after the launch of ICESat-2 have been made publicly available (https://www2.whoi.edu/site/beaufortgyre/data/mooring-data). Specifically, data from three separate moorings (A, B and D, locations shown in Figure 5) are now available from August 2018 through to August/September 2021. Each individual ULS ice draft measurement is assumed to have an uncertainty of < 10 cm (Krishfield et al., 2014). Ice draft measurements benefit from measuring a more significant fraction of the total thickness compared to freeboard, meaning they benefit from providing a reliable indicator of variability in ice thickness and avoid the complexities of snow depth retrievals. BGEP ULS data have been well utilized historically by the CryoSat-2 thickness product developers for validation, and here we offer a first assessments of ICESat-2 derived thickness data using this same data (albeit over a different time period). We follow a similar approach to other studies (Tilling et al., 2018; Landy et al., 2022) and take the mean of ICESat-2 grid-cells within a certain radius of a given mooring (we use 50 km in this study, Tilling et al., 2018 use 100 km while Landy et al., 2022 use 150 km), then find the nearest grid-cell to each mooring for a given month and compare this to a simple monthly average of all ULS data within that month. We convert the gridded ICESat-2 thickness estimates (IS2SITMOGR4 version 2) to ice draft by adding the NESOSIM snow depth and subtracting the total (ATL10) freeboard.

## 2.7 ERA5

To understand the possible relationships between seasonal/interannual differences in ice thickness and winter Arctic atmospheric conditions, we utilize near-surface (2 m) air temperature and downwelling longwave radiation estimates from the ERA5 reanalysis (Hersbach et al., 2020). A warm bias of 1-4 °C in 2 m air temperatures over Arctic sea ice in ERA5 has been noted in comparisons with drifting buoys (Wang et al., 2019; Yu et al., 2021) which has been linked to the basic (fixed thickness) representation of sea ice and its overlying snow cover in reanalyses including ERA5 (Batrak and Müller, 2019). We utilize these data with caution and focus primarily on basin-average/seasonal differences.

## 3 Results

### 3.1 ATL10 freeboards, NESOSIM snow loading and sea ice thickness distributions

In Figure 6 we show probability distributions of winter Arctic freeboards calculated using rel002 (as used in P2020) through
to rel005 ATL10 data. We show distributions from November 2018, January 2019, and April 2019 to assess differences
during different regimes of winter Arctic sea ice (early, middle and late winter) for the first season of data collection. The
distributions use data collected by beam 1 (strong) within an Inner Arctic Ocean domain (Central Arctic, Beaufort Sea,
Chukchi Sea, Laptev Sea, Kara Sea). Note that in these distributions we show only positive values of freeboard and later
snow depth and thickness, while in the raw along-track IS2SITDAT4 dataset zero freeboards, typically from open water lead
segments, are included also.

Figure 6 shows that the only notable change in freeboard distribution occurs between rel002 and rel003 - a
freeboard decrease of 1.2 cm in November 2018 (26.2 cm to 25.0 cm), a 1.9 cm increase in January 2019 (27.7 cm to 29.6
cm) and a 3.4 cm increase in April 2019 (35.9 cm to 39.3 cm). In contrast, the rel003 to rel005 freeboard distribution
differences across these three months are small or negligible (< 0.4 cm), with the rel005 freeboards generally the highest
from the four releases. As discussed earlier, this was largely expected due to the major algorithm change in rel003 (Kwok et
al., 2021d) and the lack of major algorithm changes related to freeboard derivation in rel004 and rel005. The 1.2 cm mean
freeboard reduction in November 2018 is due to a stronger primary freeboard peak and a weaker secondary peak in the
rel003 to rel005 freeboard distributions, while January 2019 and April 2019 distributions exhibit a clear increase in the
unimodal freeboard in rel003 onwards. Kwok et al., (2021d) analysed gridded freeboard distributions in January 2019, June
2019 and October 2019 and found 3 cm, 1 cm and 2 cm increases respectively between rel002 and rel003, broadly in-line
with the magnitude of the differences observed here. As discussed in Section 2.1.1 and demonstrated in Figure 2, the
different releases also include changes in coverage, especially between rel002 and rel003, which may influence these
differences along with changes in the freeboard determination algorithm.

Figure 6 (right column) also shows an analysis of the inter-beam differences across the three strong beams for the
same time periods and Inner Arctic Ocean region for rel005 data only. The inter-beam differences are small (<1 cm), similar
to the rel003 to rel005 differences. Each strong beam is separated by ~3 km across track, so greater differences are expected
at local scales, however an in-depth analysis of spatial length-scales is beyond the scope of this study. We instead note that at
basin/monthly scales, the beams provide similar freeboard distributions despite the small differences in coverage (as
discussed in Section 2.1.1 and highlighted in Figure S2), increasing our confidence in using all three beams to extend
coverage across the Arctic.

Figure 7 shows the impact of ATL10 release changes and the two NESOSIM versions (v1.0 and v1.1) on the
redistributed snow depths and resultant sea ice thickness estimates. The piecewise redistribution of the coarse 100 km
NESOSIM output to the high-resolution ATL10 data is presented in P2020 and summarized in Section 2.3. These
assessments were carried out during the period of rel004 availability, so these are the only runs with both NESOSIM

versions processed. The difference in mean redistributed snow depth across the three months and ATL10/NESOSIM configurations (left column) is < 2.2 cm, with the biggest differences occurring in November 2018, where the Nv1.1 redistributed snow depths are generally thicker than the Nv1.0 snow depths (17.0 cm compared to 15.5 cm when using the same rel004 data) driven primarily by a small positive shift in the tail of the distribution. The rel004/Nv1.0 snow depths are thinner than the rel002/Nv1.0 snow depths (17.7 cm to 15.5 cm) highlighting the non-negligible role of underlying changes

to the freeboard data in the resultant redistributed snow depths. The January and April mean snow depths are similar across the NESOSIM versions and ATL10 releases, although slight differences in the distributions are observed, e.g., a thinner secondary January 2019 snow depth peak in Nv1.1 compared to v1.0 (~22 cm compared to ~26 cm) which holds over the different ATL10 releases.

Figure 7 (right column) shows the impact of ATL10 release and NESOSIM version differences on estimates of sea

ice thickness. For the rel004 runs, the impact on thickness from the upgrade from Nv1.0 to Nv1.1 is a decrease of 10 cm in November 2018, no change in January 2019 and a decrease of 2 cm in April 2019. In contrast, the impact on thickness of the upgrade from rel002 to rel004 ATL10 (using Nv1.0) is an increase of 6 cm in November 2018, an increase of 21 cm in January 2019 and an increase of 30 cm in April 2019. The difference between the rel004 and rel005 mean thicknesses (using Nv1.1 snow loading) are < 0 to 3 cm. In summary, the impact on thickness from the choice of NESOSIM version, is less

significant than the impact from rel002 to rel003 freeboard changes.

**3.2 Gridded Arctic sea ice thickness comparisons with regional/monthly CryoSat-2 estimates using the same snow loading.**

In Figure 8, we show the correlation coefficients, mean bias and standard deviation of differences between monthly gridded ice thicknesses derived from rel002 and rel005 ICESat-2 data and thickness estimates produced from ESAs CryoSat-2. In

these thickness comparisons we generate monthly gridded thickness estimates using the same input assumptions, i.e., mW99 for snow loading (See Section 2.2.1 and Table 2 in P2020). As in P2020 we only use beam 1 (strong) and mask all data below 0.25 m and outside of an Inner Arctic Ocean domain (Figure 5) in both datasets before producing these comparisons to simplify interpretation and avoid regions of higher uncertainty within the more peripheral seas of the Arctic. As noted earlier, by using the same input assumptions we maintain focus on the potential impact of freeboard retrieval differences on

derived sea ice thickness. The large differences observed in P2020 (ICESat-2 generally much thinner than the CryoSat-2 products despite using the same loading) was noteworthy and motivated this updated comparison.

In general, the agreement between ICESat-2 and CryoSat-2 using rel005 ATL10 freeboards is much improved compared to those based on rel002 ATL10 freeboards (shown in P2020) in terms of the correlation coefficient, mean bias and standard deviation across virtually all months and products. The statistics from all months between November 2018 and

April 2019 are shown in Figure 8. More work is needed to reconcile these datasets and assess sources of bias (as discussed more in the summary) but these results represent an encouraging initial development in terms of reducing the large biases previously noted.

### 3.3 Upgraded monthly gridded sea ice thickness data (IS2SITMOGR4 v2) and comparisons with BGEP ULS data and basin-mean CryoSat-2 and PIOMAS thickness estimates

We next focus on development of the version 2 gridded thickness product and comparisons with available data. In Figure 9 we show an example output of our updated monthly gridded ICESat-2 winter Arctic sea ice thickness product (IS2SITMOGR4 version 2, v2) for April 2021 using the latest default thickness processing configuration (rel005 ATL10 and NESOSIM v1.1 snow loading). Spatial coverage is generally high across all months despite the concerns expressed in Kwok et al., (2021d) and in Figure 2 and Figure S1, related to reduced lead/sea surface height segments and thus freeboard

determination. This was partly mitigated by our use of 3 strong beams (coverage changes were discussed in Section 2.1.1). However, there are still large regions of missing data in our monthly gridded dataset, e.g., the missing data in the Laptev/East Siberian Sea shown in Figure 9 despite the monthly CDR ice concentrations showing concentrations greater than 50% in that same region. Data drop-out is often caused by the presence of clouds and the resultant atmospheric scattering impacts on ATLAS retrievals. Our interpolated/smoothed variables of freeboard, snow depth and thickness data

(Figure 9i-k, not used in this study) do not substantially increase coverage in these regions, which was in-part by design to avoid over-extrapolation of our thickness estimates. In general, the monthly gridded data gaps are limited but should be considered when using these data to assess regional and basin-scale thickness variability.

       In Figure 10 we show a comparison of IS2SITMOGR4 v2 converted to ice draft with ice draft estimates obtained by Beaufort Gyre Exploration Project (BGEP) upward looking sonar moorings (data and methodology described in Section

2.6). The comparison statistics are strong, including high correlations (squared Pearson correlation coefficient, $r^2 = 0.87$), relatively low mean bias (11 cm) and low standard deviation of differences (20 cm) when analyzing monthly mean data from all three moorings. Mooring A show higher $r^2$ than mooring B and D (0.93 for A, 0.89 for B and 0.88 for D) but these are all still considered strong. Mooring B shows significantly lower mean bias (1 cm), compared to moorings A (14 cm) and D (20 cm). As noted in Section 2.6 all IS2SITMOGR4 data with 50 km are averaged before undertaking these comparisons,

following earlier CryoSat-2 studies. In Figure S5 we also show comparisons using an averaging radius of 25 km, 50 km, 100 km and 150 km which all largely show similar results to those presented in Figure 10 (e.g., some improved correlation but also increased mean differences at 100 km). The BGEP ULS comparisons provide crucial validation of our ICESat-2 thickness data for tracking seasonal/interannual winter Arctic thickness changes.

       In Figure 11 we show a comparison of basin-averaged Inner Arctic Ocean sea ice thickness estimates between

IS2SITMOGR4 v2, several CryoSat-2 products and PIOMAS. The goal of this analysis is to provide a basic intercomparison of these commonly used datasets for inferring seasonal/interannual changes in winter Arctic sea ice thickness. Figure 11 includes the raw monthly data (within our Inner Arctic Domain) but also data on a common mask (CM) where data is masked in grid-cells missing from the other datasets. These enable a more direct comparison, while differences between the CM and non-CM results are useful indicators of the seasonal impact of spatial sampling biases when assessing basin-scale

means. In general, the agreement between IS2SITMOGR4 and the various CryoSat-2 products is strong, albeit significant differences are still observed depending on the product analysed. The AWI/SMOS comparison shows the highest correlation

($r^2 = 0.89$) and lowest mean bias (-1 cm) and standard deviation of differences (11 cm) of the CryoSat-2 products. The CPOM and UBRIS comparisons are also strong, although the CPOM thicknesses are noticeably higher (20 cm mean bias). The UBRIS comparisons are notable for showing consistently thinner ice compared to IS2SITMOGR4 prior to April, although only two winter seasons are available for this comparison. The ICESat-2/CryoSat (KK) compare the best with IS2SITMOGR4 v2, although as these are generated from ATL10 total freeboards, these are not truly independent (the strong agreement is still encouraging). The GSFC product shows the weakest agreement with IS2SITMOGR4 in terms of the squared correlation coefficient ($r^2$), mean bias and standard deviation of differences. Additional spatial difference plots are generated and provided in the online Jupyter Book.

In general, the PIOMAS results are consistent with the IS2SITMOGR4 v2 and CryoSat-2 products, although they tend to show weaker correlations compared to the CryoSat-2 comparisons and exhibit stronger seasonality than the satellite-derived products, including 20-40 cm thicker ice by the end of the season compared to IS2SITMOGR4. Additional spatial difference plots are generated and provided in the Jupyter Book which highlight the stronger disagreement at regional scales, e.g., PIOMAS not simulating the thicker ice north of the Greenland and CAA coasts shown in IS2SITMOGR4 data. These results are broadly in-line with previous published comparisons between PIOMAS and satellite altimetry-derived thickness data, including comparisons with the original ICESat mission (Schweiger et al., 2011; Petty et al., 2018b; Wang et al., 2016). The strong differences in September/October between the CM and non-CM data highlight the impact of spatial sampling issues earlier in the season (where there is a greater fraction of thin/low concentration ice in the Inner Arctic Ocean).

In summary, while there is not perfect agreement across the products in terms of the seasonal changes in winter Arctic sea ice thickness, the IS2SITMOGR4 v2 product shows good agreement with existing products, and generally lies within the CryoSat-2 product spread (mean biases that were both positive and negative depending on the product analysed). In addition to the BGEP/ULS ice draft validation, this provides us with confidence in undertaking a deeper investigation into winter sea ice changes from this dataset.

### 3.4 Three winters of sea ice freeboard, snow depth and sea ice thickness from IS2SITMOGR4 v2

In Figure 12 we show the seasonal evolution of winter freeboard, snow depth/density and sea ice thickness from our IS2SITMOGR4 v2 monthly gridded dataset. The results shown in Figure 12 are again restricted to our Inner Arctic Ocean domain. The data within this domain in September and October is generally lower concentration (~85% on average, Figure 10e), than the proceeding months, so changes in early winter are still strongly influenced by the changing coverage of sea ice as the ice pack refreezes. Note that the concentration decline from September to October is due to changes in data coverage as regions with ice concentrations < 50% are not included in ATL10 and thus our thickness estimates. We therefore mainly focus on analysing November to April changes, the period in which we also have full monthly data available across all three winters. Extended analysis of this data, e.g., utilizing different region masks, is possible through the online Jupyter Book.

Mean monthly Inner Arctic Ocean freeboards from ATL10 show a monotonic increase from 22 cm in November 2018 to 38 cm in April 2019. Mean monthly freeboards in November 2019 are similar to 2018, but consistently lower in

subsequent months (lower by ~2-3 cm). Mean freeboards in November 2020 are notably lower than the previous two winters (20 cm mean, lower by ~2-3 cm) and are similar or lower than the 2019-2020 monthly means (lower by ~1-2 cm).

        Mean monthly Inner Arctic Ocean snow depths from the redistributed NESOSIM v1.1 output monotonically increase from 14 cm in November 2018 to 24 cm in April 2019, increasing more rapidly between January and April than between November and January. Snow depths in the 2019-2020 winter show a similar seasonal evolution but with snow
depths consistently ~ 2 cm thinner than the 2018-2019 monthly means. The 2020-2021 snow depths are similar to 2019-2020, showing thicker snow in January 2021 compared to January 2020 but thinner snow in April 2021 than April 2020 and 2019. The mean seasonal snow density evolution is similar across the three winters, with the 2020-2021 density lower in November than the previous winters but notably higher than the previous winters between February and April. Due to the crude nature of the NESOSIM density parameterization, we do not view this analysis as a reliable interannual snow density
assessment but highlight this more to understand the density variability impact on our ice thickness estimates.

        Our mean monthly estimates of Inner Arctic Ocean sea ice thickness show an increase from 1.20 m +/- 0.30 m in November 2018 to 2.0 m +/- 0.35 m in April 2019, with the monthly thickness increasing more rapidly between November and February than between February and April. The 2019-2020 monthly mean thickness evolution is similar to 2018-2019 winter, but with a lower April 2020 mean thickness of 1.90 m +/- 0.35 m compared to April 2019. The 2020-2021 mean
thicknesses are notably thinner in November through February compared to the previous two winters, ~0.9 m +/- 0.3 m in November 2020 to 1.90 m +/- 0.35 m in April 2021. Analyzing just these three winters, we observe clear differences in winter Inner Arctic Ocean thickness, with variability in the NESOSIM snow depth, and density to a lesser extent, modulating a significant component of the seasonal freeboard differences observed by ICESat-2. For example: thinner 2019-2020 snow compared to 2018-2019 mitigate the thinner freeboards and results in similar mean thickness across both winters; thinner
April 2021 snow compared to 2020 and 2019 April snow mitigates the thinner freeboard and similarly results in a similar mean thickness. It is also worth noting that the fraction of multi-year ice is inversely related to the thickness rankings – i.e., the 2018-2019 winter shows the lowest mean fraction of multi-year ice in this three-year period, but also shows the highest freeboard, snow depths and thickness. Three years is not a long enough record to establish true relationships, but the results highlight the potential pitfalls of inferring thickness from ancillary quantities such as freeboard or multi-year ice fraction if
one is interested in tracking interannual changes.

        To highlight the spatial distribution of the winter changes discussed above, Figure 13 shows maps of winter mean (November to April) freeboard, snow depth and thickness from IS2SITMOGR4 v2, while Figure 14 shows anomalies relative to the three-winter mean. The most notable feature of these maps are the positive freeboard and thickness anomalies in 2018-2019 and negative freeboard and thickness anomalies in 2020-2021 north of Greenland and the Canadian Arctic
Archipelago (CAA), the region of the Arctic where we generally expect to observe the thickest freeboard, snow depth and thickness. On more regional-scales, there are noteworthy examples of the impact of the time-varying NESOSIM snow depths on our thickness retrievals, e.g., the positive snow depth anomalies in the Laptev Sea and the Central Arctic in 2018-2019 modulate the impact of the positive ATL10 freeboard anomalies on our retrieved thickness anomalies. In contrast, the strong

negative NESOSIM snow depth anomalies in 2020-2021 modulate the negative ATL10 freeboard anomalies. The low

freeboards within the Barents Sea region and the increased potential for surface flooding in this region (Granskog et al., 2017), a process which is not currently simulated by NESOSIM, means those results should also be treated with caution (this region is mostly excluded from our Inner Arctic Ocean domain, Figure 5).

These results are broadly in-line with the analysis of joint ICESat-2/CryoSat-2 sea ice freeboard, snow depth and thickness presented in (Kacimi and Kwok, 2022). Their results, analysed within a similar Inner Arctic Ocean region show

encouraging agreement with the results presented here: (i) snow depths increasing from ~10 - 12 cm in November to ~20-22 cm in April, ~2 cm thinner than our snow depths but with a similar overall decline of ~3 cm over the three year period and (ii) thickness increasing from ~1.0 - 1.3 m in November to 2.1 - 2.4 m in April, ~10-40 cm thicker than our snow depths but with a similar overall decline of ~10-30 cm depending on the month analysed, with 2020-2021 notably thinner.

### 3.3.1 Causes of winter Arctic thickness differences

The regional anomaly maps allude to a strong ice type dependency as some of the more notable winter anomalies are observed within the thicker/older ice of the Central Arctic. To explore this further, we explore the seasonal timeseries of IS2SITMOGR4 v2 but delineated by ice type (data still masked outside the Inner Arctic Ocean domain). Figure 15 shows the mean seasonal time series of regions identified as first-year ice (FYI) only, limited to October onwards due to the lack of reliable FYI coverage data in September. The differences in FYI freeboard and snow depth in November/December are

small (<1 cm) across the three winters. The higher 2018-2019 FYI freeboard and snow depth compared to the two more recent winters (~2-4 cm higher on average) is observed later in the season compared to the 'all ice' analysis (Figure 13). The resultant FYI sea ice thickness winter timeseries comparison is notable for its consistency across the three winters (interannual thickness differences < 15 cm across all months).

Figure 15 shows the mean seasonal time series of regions identified as multiyear ice (MYI) only. The interannual

MYI differences across most variables are higher than the FYI differences. The 2018 November MYI freeboards are 5 cm higher than the 2019 and 2020 Novembers. These freeboard differences largely persist until February onwards when the 2019-2020 freeboard increases in-line with coincident increases in snow depths. The result of these interannual freeboard and snow depth differences is MYI thickness that exhibits similar seasonal cycles across the three winters but with differences of 10 to 50 cm across the years. The thickness differences largely persist across the three winters with each year

thinner than the one before and 2020-2021 winter Arctic MYI thicknesses that are ~ 50 cm thinner than the 2018-2019 winter. This 50 cm MYI thickness decline in just this three year period was also highlighted in Kacimi and Kwok, (2022). The MYI results also highlight the important role of dynamic snow loading in constraining regional/monthly thickness variability, e.g., the November-December 2019-2020 and 2020-2021 freeboards are near identical, but NESOSIM indicates 2020-2021 snow depths are ~ 5 cm thicker, resulting in a ~30 cm thicker ice estimate.

These ice type differences align with our general understanding of winter sea ice thickness and growth – thinner ice is more responsive to atmospheric forcing and can thicken rapidly due to its reduced insulation (the negative feedback of ice

growth) so small differences in the thickness of thin FYI at the start of winter are not expected to be good predictors of end-of-winter thickness (Petty et al., 2018b). Conversely, MYI is thicker at the start of winter, meaning thickness anomalies are more likely to persist through winter as the ice is more insulated and less sensitive to atmospheric forcing. The differences in

MYI thickness at the start of our three winters appears to provide a strong control on the total (combined MYI and FYI) winter thickness anomalies across all months, albeit in this limited record.  Previous studies based on CryoSat-2 derived Arctic sea ice thickness estimates have highlighted the important role of variable summer conditions in determining start of winter ice thickness anomalies and thus total winter thickness (and volume) anomalies  (Tilling et al., 2015; Kwok, 2015). More specifically, a sharp increase in the start-of-winter 2013 Arctic thickness/volume was related to reductions in the

duration of the summer melt season (Tilling et al., 2015) and also to dynamically driven convergence of ice within the Central Arctic (Kwok, 2015). The observed positive autumn 2013 thickness anomaly persisted through winter months, as in our 2018/2019 results. We do not seek to provide a similar level of analysis in this study as our primary goal was to highlight and describe this new thickness dataset, but the agreement with this prior physical understanding is encouraging.

To better understand the regional differences, the spatial thickness maps in Figure 11 include winter mean ice drifts

from the monthly OSI SAF global low resolution ice drift product (Lavergne et al., 2010). In general, the mean circulation across these three winters are similar – featuring anti-clockwise Beaufort Gyre circulations and Transpolar drifts, but with some key differences. For example, ice drifts through the southern Beaufort Sea in 2018-2019 winter were stronger than the 2019-2020 and 2020-2021 drifts, which is likely associated with the positive freeboard/snow depth/thickness anomalies observed in the Chukchi Sea and negative anomalies in the Beaufort Sea in 2018-2019. Disentangling cause from effect is

challenging as ice drift is strongly influenced by the sea ice conditions (Petty et al., 2016), however the strong association between drift patterns and thickness anomalies is again encouraging. Stronger ice drift anomalies are apparent when assessing monthly (not seasonal) differences, which can be explored more in the relevant Jupyter Book page.

Finally, to briefly explore the connection between the interannual/seasonal thickness differences and variability in atmospheric conditions, Figure S6 shows near-surface (2 m) air temperature and downwelling longwave radiation from

ERA5 averaged over our Inner Arctic Ocean domain. The 2018-2019 winter shows lower temperatures and downward longwave fluxes at the start of winter (November through January) compared to 2019-2020 and 2020-2021, but higher temperatures and downward longwave flux in the middle-end of winter (February and March) compared to the following winters. April temperatures and downward longwave are similar in 2018-20919 to 2019-2020. It has been well-established that near-surface atmospheric conditions are strongly coupled to variability in the sea ice state so, as in the ice drift analysis,

it is challenging to differentiation cause from effect. Nevertheless, our limited three-year analysis provides some limited evidence of the strong link between near-surface atmospheric conditions during the start-middle of winter and interannual winter ice conditions, promoting persistence of the interannual start-of-winter thickness anomalies (especially for the thicker MYI). A longer time-series, ideally complemented by fully coupled climate model studies, is needed to explore these relationships in more detail and the issues with ERA sea ice representation and impacts on these results (e.g. the ERA5 warm

bias, as noted in Section 2.7) means these results should be treated with caution.

## 4 Summary

In this study we provided an impact assessment of upgrades to the input data used to produce ICESat-2-derived winter Arctic sea ice thickness estimates shown in Petty et al., (2020), and an extended analysis of the upgraded monthly gridded winter Arctic thickness dataset across the three winters profiled since the launch of ICESat-2 in September 2018.

Input data upgrades include the ICESat-2 ATL10 freeboards (Release 002 to 005, rel002 to rel005) and NASA Eulerian Snow On Sea Ice Model (NESOSIM, version 1.0 to version 1.1) snow loading. A key change in ATL10 data was the removal of misclassified leads from the determination of sea surface and thus freeboard in rel003. This was thought to be the primary cause of the increase in freeboard observed in January 2019 and April 2019 in rel003 data compared to rel002, together with the stronger primary peak of lower freeboards in November 2018 rel003 data. Later releases of ATL10 (rel004
and rel005) involved only minor changes to the freeboard algorithms and thus exhibit less significant changes in the observed freeboard distributions compared to the rel002 to rel003 change. The different releases also show slight differences in ATL10 data coverage, due primarily to the changes associated with dark lead usage, but also the inclusion/filtering of satellite calibration scan data. Our updated version 2 monthly gridded winter Arctic sea ice thickness dataset now utilizes all three strong beams to help mitigate these coverage issues and includes preliminary interpolated/smoothed data variables.

The upgrades to NESOSIM (version 1.1) presented in this study includes a new wind-driven atmosphere snow loss term, CloudSat-scaled ERA5 snowfall forcing (Cabaj et al., 2020) and some more minor bug fixes. NESOSIM v1.1 was also re-calibrated (heuristically) using spring Arctic snow depth estimates obtained by NASA's Operation IceBridge airborne mission (a gridded median estimate derived in this study from available datasets). NESOSIM v1.1 generally shows similar snow depths to NESOSIM v1.0, resulting in a less significant impact on Arctic winter sea ice thickness compared to the
rel002 to rel003 freeboard changes.

The rel005-derived monthly gridded winter Arctic ice thickness data show improved comparisons with thickness estimates produced from ESA's CryoSat-2 using the same input assumptions across all 2018-2019 winter months (lower mean biases and standard deviations, higher correlations) compared to rel002-derived estimates.

This study also provided additional validation of the gridded data through comparisons with Upward Looking Sonar
moorings deployed in the Beaufort Sea through the Beaufort Gyre Exploration project (BGEP). The comparisons between our updated monthly gridded winter Arctic sea ice thickness dataset (IS2SITMOGR4 v2) and BGEP were notably strong, including an $r^2$ of 0.87, mean differences of 11 cm and standard deviation of differences of 20 cm. The strength of the agreement generally held across the three different moorings and for different averaging length scales. The results of this validation analysis are at least comparable (and generally better) than similar validation efforts reported from CryoSat-
2/BGEP studies, however these likely depend strongly on the chosen time range and comparison methodology.

We also showed comparisons of basin-mean monthly winter Arctic sea ice thickness from several CryoSat-2 products (and a merged CryoSat-2/ICESat-2 product) and PIOMAS to provide a basic intercomparison of these commonly used thickness datasets for tracking seasonal/winter thickness changes. Generally, the agreement between IS2SITMOGR4 v2

and the CryoSat-2 products was high, especially for the AWI/SMOS, CPOM and UBRIS CryoSat-2 products and especially the (not-independent) merged ICESat-2/CryoSat-2 derived thickness product. No consistent biases were found between IS2SITMOGR4 v2 and the CryoSat-2 products (mean biases were both positive and negative depending on the product analyzed). Agreement between IS2SITMOGR4 v2 and PIOMAS was good, although generally not as strong as the comparisons to the CryoSat-2 products. More significant differences were noted between ICESat-2 and PIOMAS at more regional scales, e.g., the lack of thick ice along the Canadian/Greenland coast.

Finally, we presented estimates of winter Arctic sea ice thickness from this over the past three winter seasons of data collection (November 2018 – April 2021, September 2019 - April 2020 and September 2020 – April 2021). Our results showed clear differences in mean winter Arctic sea ice thickness within our Inner Arctic Ocean domain across the three winters profiled, due primarily to differences in the multiyear ice thickness across the three winters (multiyear ice thinning of 10 to 50 cm each year across the three winters analysed). Interannual changes in snow depth provide significant regional/monthly impacts on our thickness results – mitigating some, or in some cases all, of the impact from interannual differences in Arctic winter freeboards observed by ICESat-2.  These results provide further evidence of the important role of dynamic snow loading when assessing interannual variability in winter Arctic sea ice thickness from satellite altimetry (Bunzel et al., 2018; Mallett et al., 2021). Specific regional thickness anomalies, e.g., in the Southern Beaufort and Chukchi seas, were also associated with interannual ice drift anomalies.

## 4.1 Future work

**ICESat-2/ATL10**: Work is still on-going to re-introduce dark leads to the sea surface and freeboard algorithm in ATL10, which requires a new filter to skilfully discriminate dark lead segments (low photon rate) from segments with photon attenuation driven by the presence of clouds. The variable properties of clouds and their impact on photon attenuation, together with the limited availability of coincident imagery for validation (as used in Petty et al., 2021) makes this development challenging.  An additional near-term goal related to ATL10 is the plan to utilize all six beams, or at least the three strong beams, concurrently to produce two-dimensional interpolated fields of sea surface height, as opposed to the independent beam processing currently utilized.  However, residual absolute height biases of several centimetres are still observed between the beams as of Release 005 (updated from the analysis by Bagnardi et al., 2021, not shown), hindering this development. More sophisticated sea surface interpolation methods should also be explored (Landy et al., 2021). Algorithm development efforts related to these issues are on-going through the ICESat-2 Project Science Office to be included in future ATL10 data releases.

**Snow loading**: Work is on-going to utilize a Markov Chain Monte Carlo (MCMC) approach to automate the calibration of NESOSIM and provide a more robust uncertainty estimate of snow depth and density from this simple model framework (Cabaj et al., 2021). This approach benefits from the low computational cost of NESOSIM, allowing thousands of model simulations to be generated across plausible model parameter space. Additional physical upgrades are still desired,

e.g., the introduction of a snow melt parameterization to extend NESOSIM through summer. However, additional reliable ground-truth data at regional/basin-scales are needed to calibrate and validate such development activities.

Recent studies leveraging newly generated Arctic snow reconstructions and satellite-derived data products, including the joint ICESat-2/CryoSat-2 derived snow depths, are helping collectively provide new insights into snow depth variability and its impacts on sea ice thickness and its contribution to total thickness uncertainty (Zhou et al., 2021; Mallett et al., 2021; Glissenaar et al., 2021; Kacimi and Kwok, 2022). While these datasets, including NESOSIM, are still generally limited by a lack of contemporary ground-truth data for assessing data accuracy, the creation of new operational, i.e., continuously updated and disseminated, snow products should help enable more comprehensive assessments of systematic snow loading uncertainties. The comprehensive in-situ snow observations collected from recent campaigns, including the Multidisciplinary drifting Observatory for the Study of Arctic Climate (MOSAiC) expedition (Wagner et al., 2022) can hopefully aid with the continued refinement of these new snow reconstructions and redistribution methods.

**ICESat-2-derived sea ice thickness:** Our primary focus of the three-winter thickness assessment was the monthly gridded winter Arctic thickness dataset (IS2SITMOGR4 v2, Petty et al., 2022b). We hope to incorporate a more comprehensive and accurate accounting of the various error contributions where possible, e.g., accounting for the clear representation error associated with grid-cell sampling time differences, while also exploring more sophisticated uncertainty quantification methods (e.g. Monte Carlo approaches). Raw (and 10 km smoothed) along-track data at the segment resolution of ATL10 (~20 m) are also available (IS2SITDAT4, Petty et al., 2022a, data shown in Figure 7), which provide higher fidelity information regarding the sea ice state than the monthly gridded estimates. Efforts have been recently undertaken to assess the winter Arctic sea ice thickness distribution from these data including comparisons with model-based estimates (Smith et al., 2022). Continued refinement and/or redevelopment of the snow redistribution scheme is expected. We also hope to combine these data with new ICESat-2-derived floe size estimates (Petty et al., 2021) towards a joint floe size-thickness distribution in combination with efforts to improve the accuracy of the lead/ice discrimination. The along-track dataset is more computationally demanding but increasing access to high performance computing environments (e.g., cloud compute platforms) would help increase its usability. The extension of NESOSIM through summer months will help enable summer preliminary production of summer Arctic thickness estimates, together with improved understanding of the performance of ICESat-2 over the complex summer melt surface for summer freeboard determination (Tilling et al., 2020). A recently completed 2022 ICESat-2 summer airborne cal/val campaign should also provide important insights towards this goal.

**Sea ice thickness reconciliation**: The improved correspondence between our ICESat-2 derived estimates of winter Arctic sea ice thickness and those generated from ESA's CryoSat-2 are encouraging. There are clear advantages (and disadvantages) from estimating sea ice thickness from either radar or laser altimetry, which need to be better considered and utilized for constraining total Arctic, and eventually Antarctic, sea ice volume. Radar altimeters, e.g., CryoSat-2, are highly sensitive to leads and are unaffected by clouds, providing benefits to both the quality and coverage of data collected. In contrast, laser altimeters (e.g., ICESat/ICESat-2) generally provide higher resolution data and obtain more precise estimates

of the snow-covered ice surface height (and thus total freeboard) compared to the arguably less distinct/certain ice-snow interface height (and thus ice freeboard) obtained by typical radar altimeters. The effective radar penetration depth at *Ku*-band is generally considered to come from the ice-snow interface although recent studies continue to challenge this (Nandan et al., 2017, King et al., 2018). As ICESat-2 profiles the upper snow surface, there is also more constraint on the total snow loading (it cannot be more than the measured freeboard). In both cases, uncertainties in the derived freeboard estimates are combined with uncertainties in the various input assumptions (snow loading, sea ice density) to provide total thickness uncertainty estimates. Constraining the various input uncertainties and residual biases remains challenging, which points to the need for improved exploitation of existing ground-truth data and further field and airborne campaigns considering the fast-changing Arctic. The good agreement between our results and the various CryoSat-2 products on a basin-averaged scale is also encouraging, but significant differences are still observed across specific months and at more regional scales that is worthy of further investigation. Improvements to the underlying freeboard algorithms and input assumptions are urgently needed as we seek to reconcile these datasets and hopefully move towards multi-sensor thickness assessments (increasing coverage and data quality). More work is also needed to agree on standardized methods of uncertainty quantification and propagation of errors for both along-track and gridded datasets, accounting for the different spatial scales involved. Planning is underway for a coordinated intercomparison exercise around new semi-synchronous along-track measurements available since the CRYO2ICE orbit alignment (https://earth.esa.int/eogateway/missions/cryosat/cryo2ice).

**Code availability**

Our analysis of the monthly gridded winter Arctic thickness data (IS2SITMOGR4) described above (Figures 9-16) have been summarized and made available through an online Jupyter Book (https://www.icesat-2-sea-ice-state.info). The Jupyter Book consists of a series of Jupyter Notebooks that provide all code and analysis output written in the Open-Source Python programming language for demonstrating and sharing our thickness analysis workflow. The development of the Jupyter Book was motivated by the desire for transparency and the broader goals of facilitating more open science, but also the desire to provide a simple mechanism for interested users to explore regions and time periods not shown here. For example, the Jupyter Book allows users to adapt the code interactively, either locally or using Binder (https://mybinder.org), to select months and regions of interest to explore characteristics of this dataset beyond the core figures we show here. It is our expectation that this Jupyter Book will be updated as new IS2SITMOGR4 data (and ideally the IS2SITDAT4 along track data product) are created and made public to enable continued assessments of winter Arctic thickness change. A version tagged version of this Jupyter Book will be obtained and archived on Zenodo on completion of peer review.

NESOSIM is available on GitHub (https://github.com/akpetty/NESOSIM/) and the version 1.1 release used in this study has been tagged as a specific release on GitHub and archived on Zenodo (https://doi.org/10.5281/zenodo.4448356).

The original sea ice thickness processing code presented in Petty et al., (2020) is available on GitHub (https://github.com/akpetty/ICESat-2-sea-ice-thickness, all in the open-source language Python). We plan to update this using the small upgrades made to our processing chain on completion of this peer-review.

**Data availability**

The monthly gridded winter Arctic sea ice thickness data derived in this study (IS2SITMOGR4, version 2) is being made
available through the National Snow and Ice Data Center (NSIDC) (https://nsidc.org/data/IS2SITMOGR4, Petty et al., 2022b). The along-track (raw and 10 km mean) Arctic sea ice thickness estimates are also in the process of being ingested and made publicly available through the NSIDC (IS2SITDAT4, https://nsidc.org/data/IS2SITDAT4, Petty et al., 2022a).

        The ICESat-2 ATL10 sea ice freeboard data (currently Release 005) can be obtained from the NSIDC (https://nsidc.org/data/atl10). NSIDC generally maintains an archive of ICESat-2 data from the current and previous release,
so currently Release 004 can be obtained from the NSIDC also (https://nsidc.org/data/atl10/versions/4).

        The output from our NESOSIM v1.1 model framework from 1980-2021 and the NESOSIM v1.1 climatology presented here has been archived on Zenodo (https://doi.org/10.5281/zenodo.5164314).

        Daily and monthly NASA Climate Data Record (CDR) version 4 ice concentration data were obtained from the NSIDC (https://nsidc.org/data/G02202). ERA5 estimates of daily snowfall, winds and near surface temperature and
downwelling longwave radiation were obtained from the European Centre for Medium-Range Weather Forecasts (ECMWF) Copernicus Climate Change Service Climate Data Store (https://cds.climate.copernicus.eu). EUMETSAT OSI SAF ice motion data were obtained through their web portal (http://osisaf.met.no/p/ice/, last access: 1 May 2021). OSI SAF ice type data were obtained from their ftp repository (ftp://osisaf.met.no/prod/ice/type/, last access 1 May 2021). Polar Pathfinder version 4 ice drifts were obtained from the NSIDC (https://nsidc.org/data/nsidc-0116/versions/4).

The NASA GSFC CryoSat-2 (CS-2) Arctic sea ice thickness data were obtained from the NSIDC (https://nsidc. org/data/RDEFT4, last access: 1 May 2019). The CPOM CS-2 thickness data were obtained from their web portal (http://www. cpom.ucl.ac.uk/csopr/seaice.html, last access: 1 May 2019). The AWI CS-2 thickness data were obtained from their web portal (http://data.seaiceportal.de/data/cryosat2/version2.1/l3c_grid, last access: 1 May 2019). The NASA JPL CS-2 thickness data were obtained directly from Dr. Ron Kwok. The AWI/SMOS CS-2 thickness data were obtained from their
web portal (https://spaces.awi.de/pages/viewpage.action?pageId=291898639, last access: 1 September 2022). The UBRIS CS-2 thickness data were obtained from the British Antarctic Survey Polar Data Centre (https://doi.org/10.5285/D8C66670-57AD-44FC-8FEF-942A46734ECB). The merged CS-2/IS-2 thickness data were obtained from the ICESat-2 website (https://icesat-2.gsfc.nasa.gov/sea-ice-data/kacimi-kwok-2022). The 2018-2021 BGEP/ULS mooring draft data were obtained from their web portal (https://www2.whoi.edu/site/beaufortgyre/data/mooring-data/2018-2021-mooring-data-from-
the-bgep-project/).

## Author contributions

AP led the study, produced the updated NESOSIM v1.1 framework/output and ICESat-2 thickness estimates (along-track and gridded). NK produced the Jupyter Book with assistance from AP and helped produce the new interpolated thickness variables. AC and PK worked with AP to produce the updated NESOSIM v1.1. framework and output. MB generated the release and beam coverage assessments and provided key input on ATL10 upgrades. AP wrote the manuscript with input/edits provided from all authors.

## Competing interests

The authors declare that they have no conflict of interest.

## Acknowledgements

We would like to thank the ICESat-2 scientists and engineers for their continued efforts in enabling the production of the high-quality freeboard data analysed in this study. Thanks also to the NSIDC team for their continued support in documenting and hosting our derived winter Arctic sea ice thickness data.

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

| Model parameter | NESOSIM v1.0 | NESOSIM v1.1 |
|---|---|---|
| New snow density, top layer (kg m$^{-3}$) | 200 | 200 |
| Old snow density, bottom layer (kg m$^{-3}$) | 350 | 350 |
| Wind action threshold (m s$^{-1}$), $\alpha$ | 5 | 5 |
| Blowing snow open water loss coefficient (s$^{-1}$), $\beta$ | 2.9 x 10$^{-7}$ | 1.45 x 10$^{-7}$ |
| Blowing snow atmosphere loss coefficient (s$^{-1}$), $\gamma$ | N/A | 2 x 10$^{-8}$ |
| Wind packing coefficient, $\omega$ (s$^{-1}$) | 5.8 x 10$^{-7}$ | 5.8 x 10$^{-7}$ |
| **Forcing data** | | |
| Snowfall | MEDIAN-SF (Sep 2000 to Apr 2015) ERA-I (Sep 2018 to Apr 2019) | ERA5 (+ CloudSat scaling) |
| Near-surface winds | ERA-I | ERA5 |
| Near-surface air temperature | ERA-I | ERA5 |
| Sea ice concentration | Bootstrap (Sep 2000 to Apr 2015) NSIDC CDRv3 (Sep 2018-Apr 2019) | NSIDC CDR v3 |
| Sea ice drift | NSIDC v3 (Sep 2000-Apr 2015) OSI SAF (Sep 2018-Apr 2019) | NSIDC v4 (Sep 1980 to April 2019) OSI SAF (Sep 2019 to Apr 2021) |
| **Initial conditions** | | |
| Start date | August 15th | September 1st |

**Table 1:** Model configurations for NESOSIM v1.0 and v1.1.

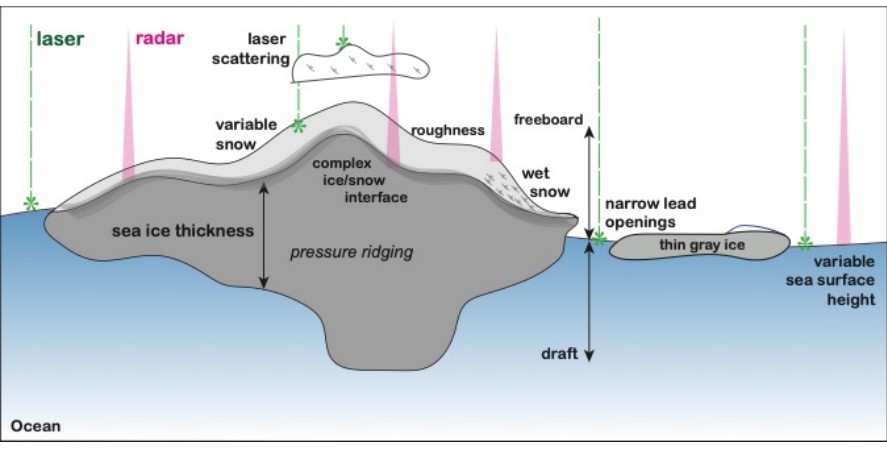

**Figure 1:** Schematic showing the typical approach and key challenges in active sea ice altimetry (laser, e.g., ICESat-2, radar, e.g., CryoSat-2) over winter sea ice. Not to scale.

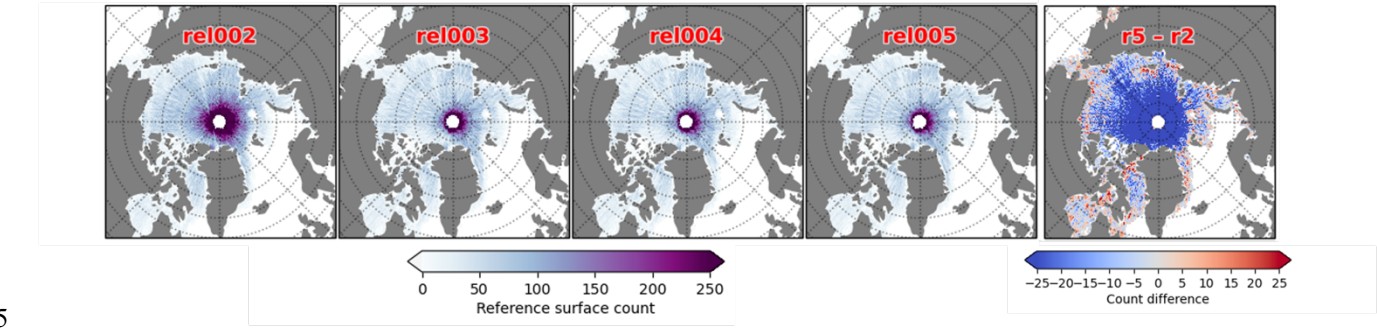

**Figure 2:** number of 10 km along-track reference sea surfaces from the three strong beams from November 2018 to April 2019 for Release 002/rel002 (left) through to Release 005/rel005 (4th from left), then (right) difference in reference surface counts between rel005 relative to rel002.

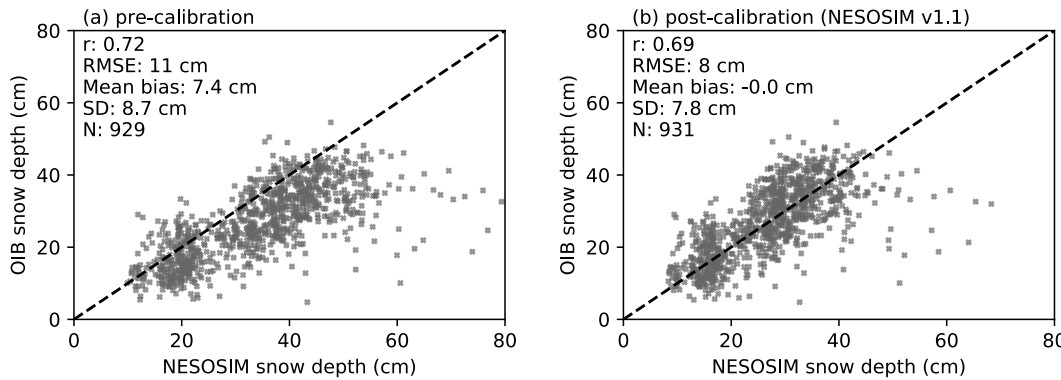

**Figure 3**: Comparison of (a) pre-calibration NESOSIM v1.1 and (b) post-calibration, NESOSIM v1.1 snow depths against spring (2010-2015) Arctic snow depths from gridded daily spring 2010 to 2015 median Operation IceBridge (OIB) snow depth estimates.

1055

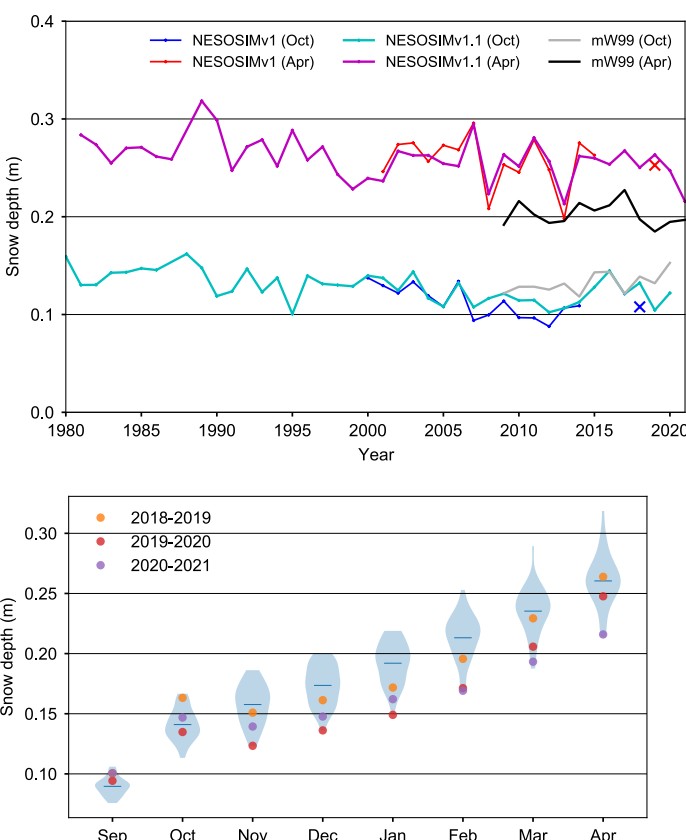

**Figure 4**: (top) Mean snow depths in October and April from NESOSIM v1.0 (blue and red respectively) and v1.1 (cyan and magenta) within an Inner Arctic Ocean domain (Figure 5). NESOSIM snow depths are also masked where concentration (from passive microwave) is less than 50%. The cross markers show the extended ICESat-2 NESOSIM v1.0 results used in (Petty et al., 2020). The black/gray lines show the modified Warren climatology (mW99) in October and April respectively for regions of coincident NESOSIM v1.1 coverage in that given year. (bottom) violin plots showing interannual distributions of monthly mean snow depths from NESOSIM v1.1 within an Inner Arctic Ocean domain from 1980-2021, colored markers indicate mean monthly snow depths for recent (ICESat-2) winters.

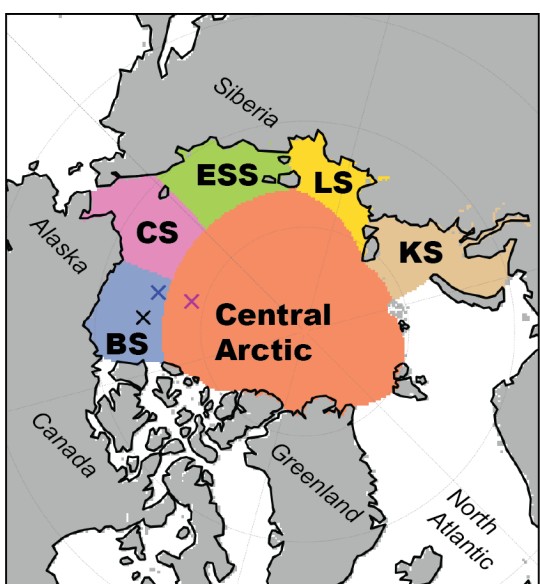

**Figure 5**: Inner Arctic Ocean study domain, defined as the combined area of the Central Arctic, Beaufort Sea (BS), Chukchi Sea (CS), E Siberian Sea (ESS), Laptev Sea (LS) and Kara Sea (KS), adopted from a region mask of the Arctic Ocean from the National Snow and Ice Data Center (NSIDC) provided by W. Meier & S. Stewart. The blue, magenta and black crosses mark the location of BGEP Moorings A, B and D.

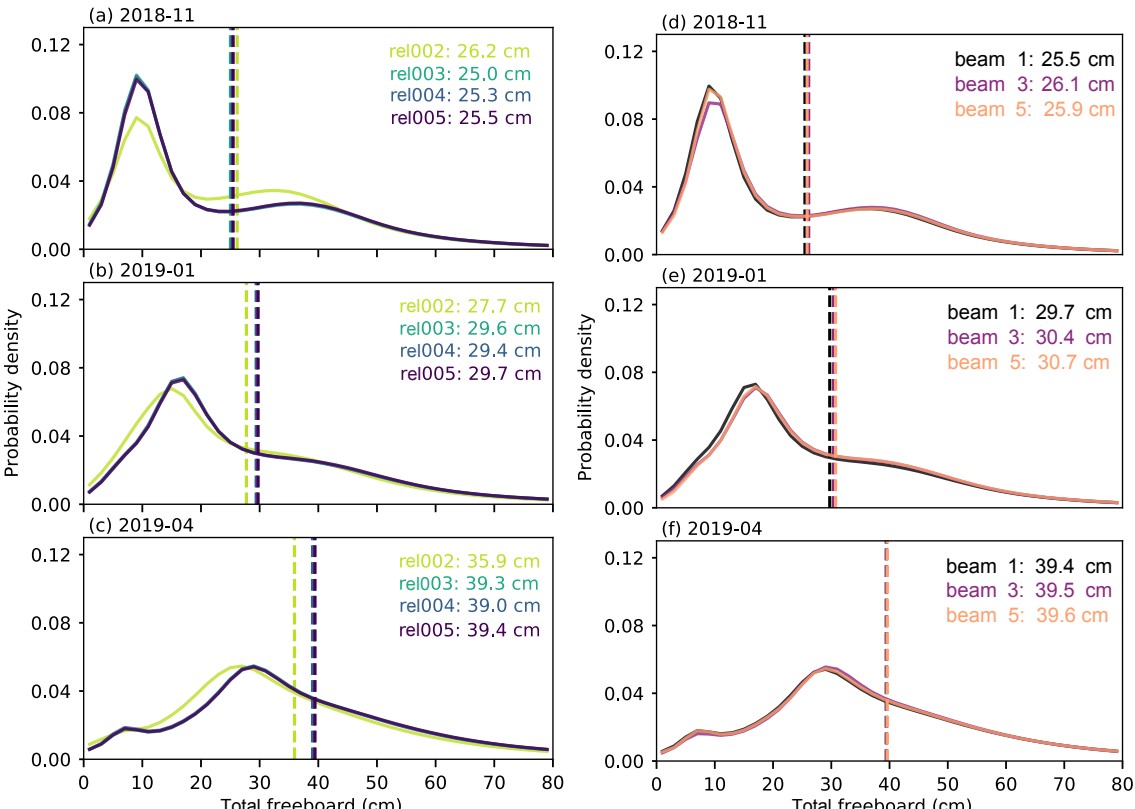

**Figure 6:** Probability distributions in November 2018 (top row), January 2019 (middle) and April 2019 (bottom) of ATL10 total freeboard for (left column) Release 002 (rel002) to Release 005 (rel005) using beam 1 (strong) and (right column) strong beam 1 3 and 5 (all three strong beams) for Release 005 (rel005) data. All distributions only show data collected within an Inner Arctic Ocean domain (Figure 5). Dashed lines show the mean values of each distribution.

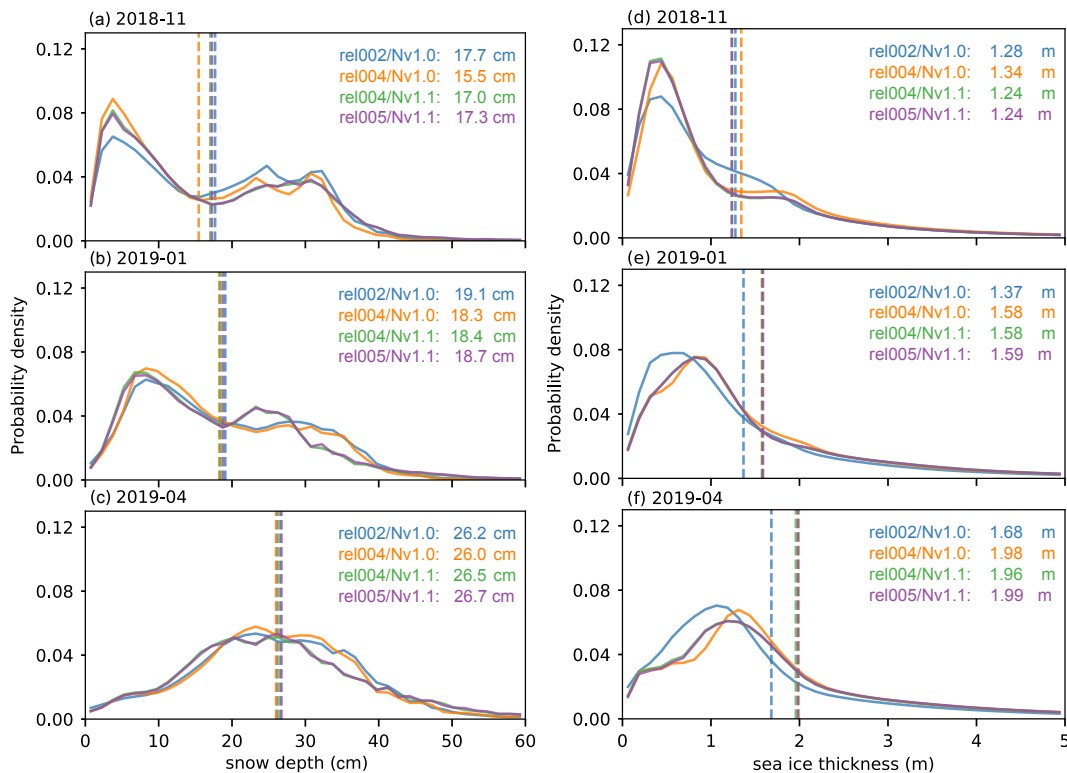

**Figure 7:** Probability distributions in November 2018 (top), January 2019 (middle) and April 2019 (bottom) using different combinations of ATL10 release and snow depth/density from NESOSIM v1.0 (Nv1.0) and v1.1 (Nv1.1) of (left column) redistributed snow depth and (right column) resultant sea ice thickness. All distributions only show data collected within an Inner Arctic Ocean domain (Figure 5) using data from beam 1 (strong). Dashed lines show the mean values of each distribution.

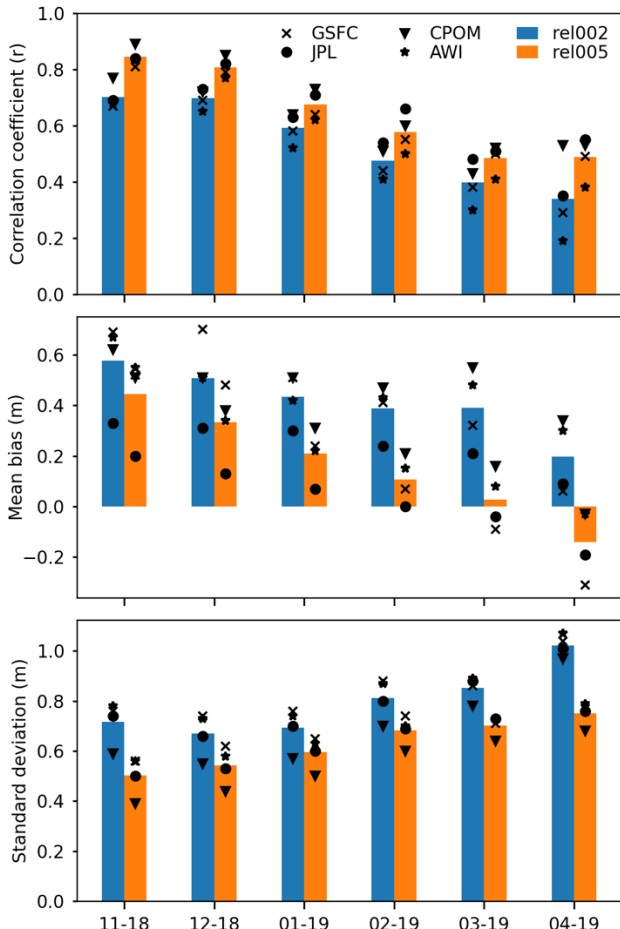

**Figure 8:** Comparison statistics of monthly gridded CryoSat-2 thickness for four different CryoSat-2 products (GSFC, JPL, CPOM, AWI) with monthly gridded ICESat-2 sea ice thickness using rel002 (blue) and rel005 (orange) ATL10 and the same snow loading and ice density input assumptions from November 2018 (11-18) to April 2019 (04-19). Data are compared within our Inner Arctic Ocean domain and for grid-cells in both datasets that contain thicknesses > 0.25 m.

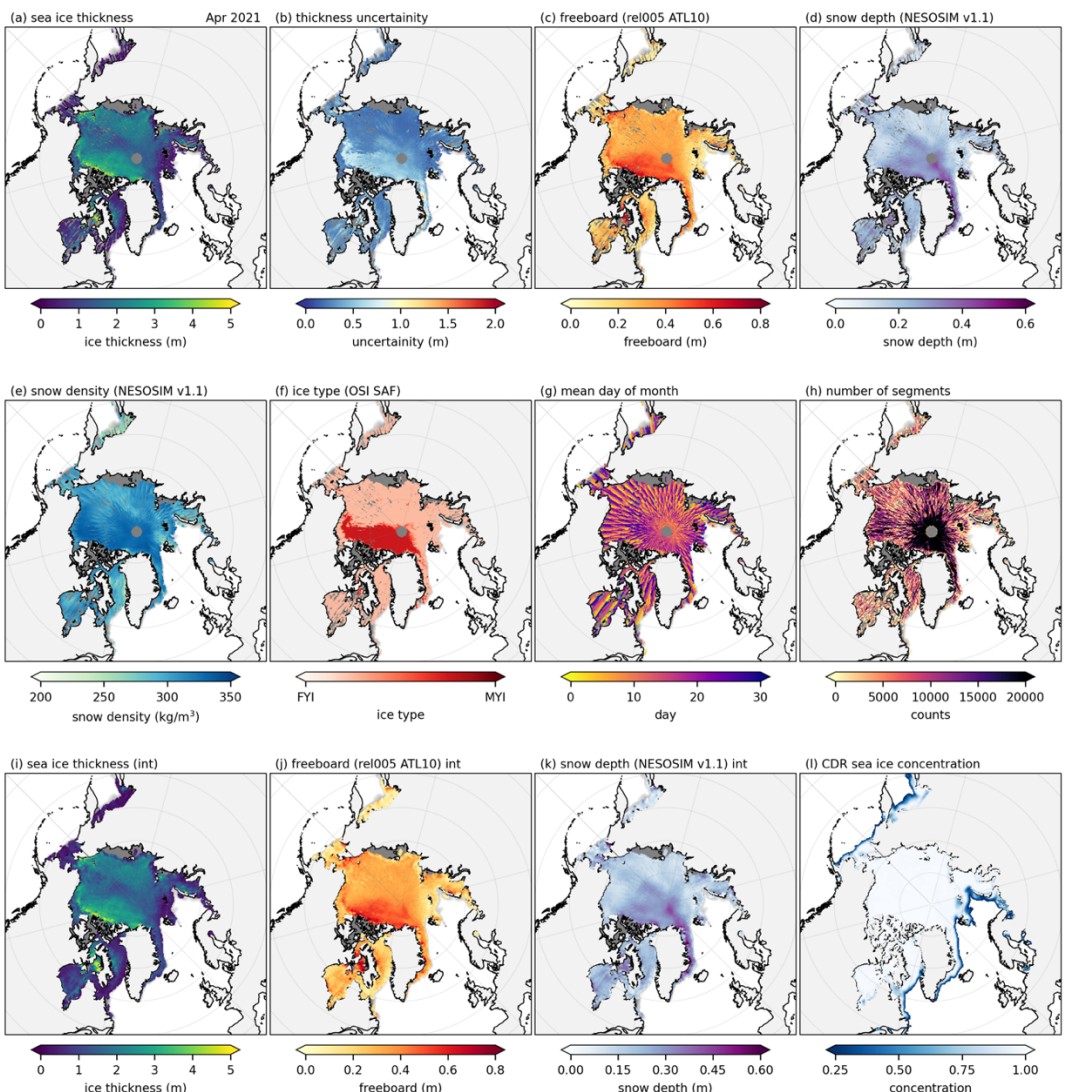

**Figure 9:** Example monthly mean gridded sea ice thickness dataset (IS2SITMOGR4, version 2) for April 2021. Dataset derived from rel005 ATL10 freeboards and NESOSIM v1.1 snow loading across all three strong beams. The background dark gray shading in panels a to k is the CDR sea ice concentration shown in panel l. Panels i to k show interpolated/smoothed variables.

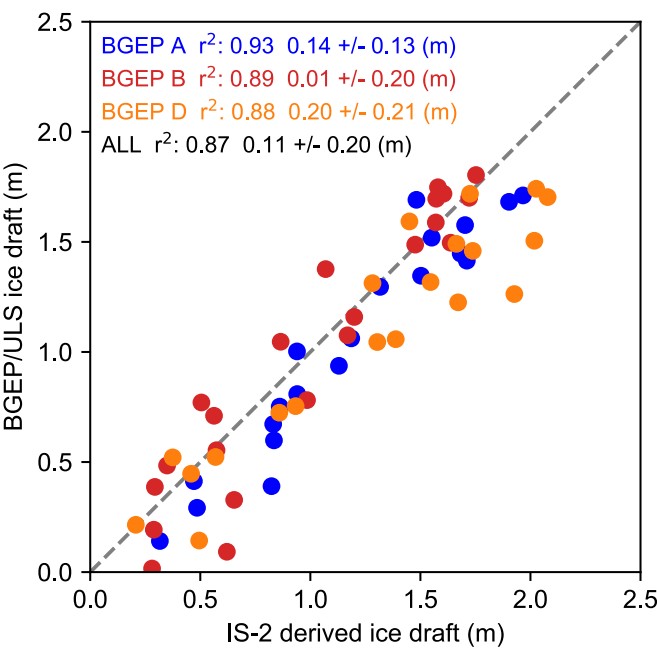

BGEP A  $r^2$: 0.93  0.14 +/- 0.13 (m)
BGEP B  $r^2$: 0.89  0.01 +/- 0.20 (m)
BGEP D  $r^2$: 0.88  0.20 +/- 0.21 (m)
ALL  $r^2$: 0.87  0.11 +/- 0.20 (m)

**Figure 10:** Comparisons of IS2SITMOGR4, v2 (Nov 2018-April 2019, September 2019-April 2020, September 2020-April 2021) converted to ice draft against ice draft measurements obtained by Beaufort Gyre Exploration Project (BGEP) upward looking sonar moorings. The mean of all IS2SITMOGR4 data within 50 km of the given mooring are used in this comparison.

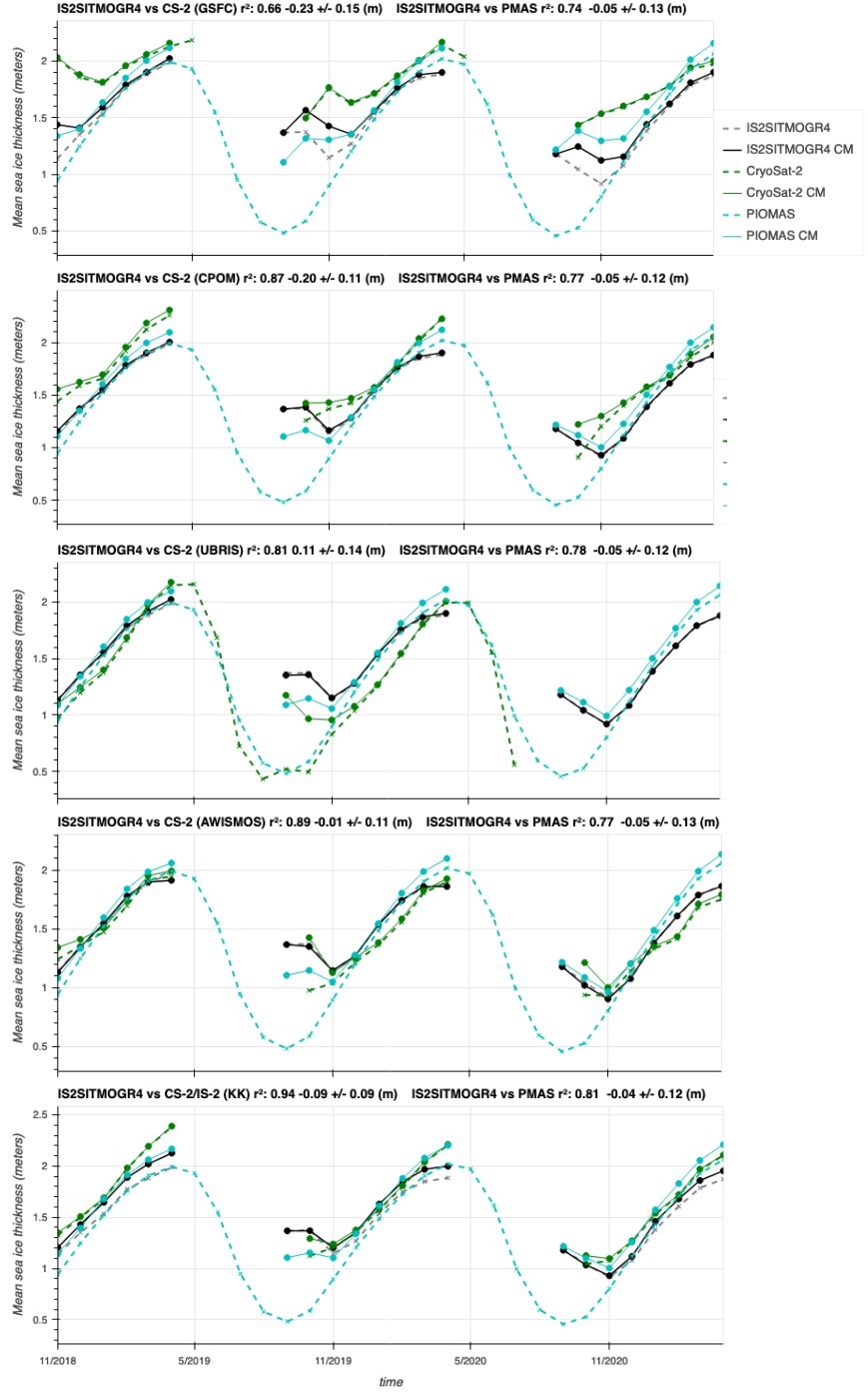

**Figure 11:** Comparisons of IS2SITMOGR4, v002 (ICESat-2) against thickness estimates from PIOMAS and various CryoSat-2 thickness products. CM: mean calculated using a common spatial grid-cell mask in the given month. Correlation

coefficient, mean bias and standard deviation of differences between IS2SITMOGR4CM and CryoSat-2 CM and IS2SITMOGR4 and PIOMAS are given in each sub-heading.

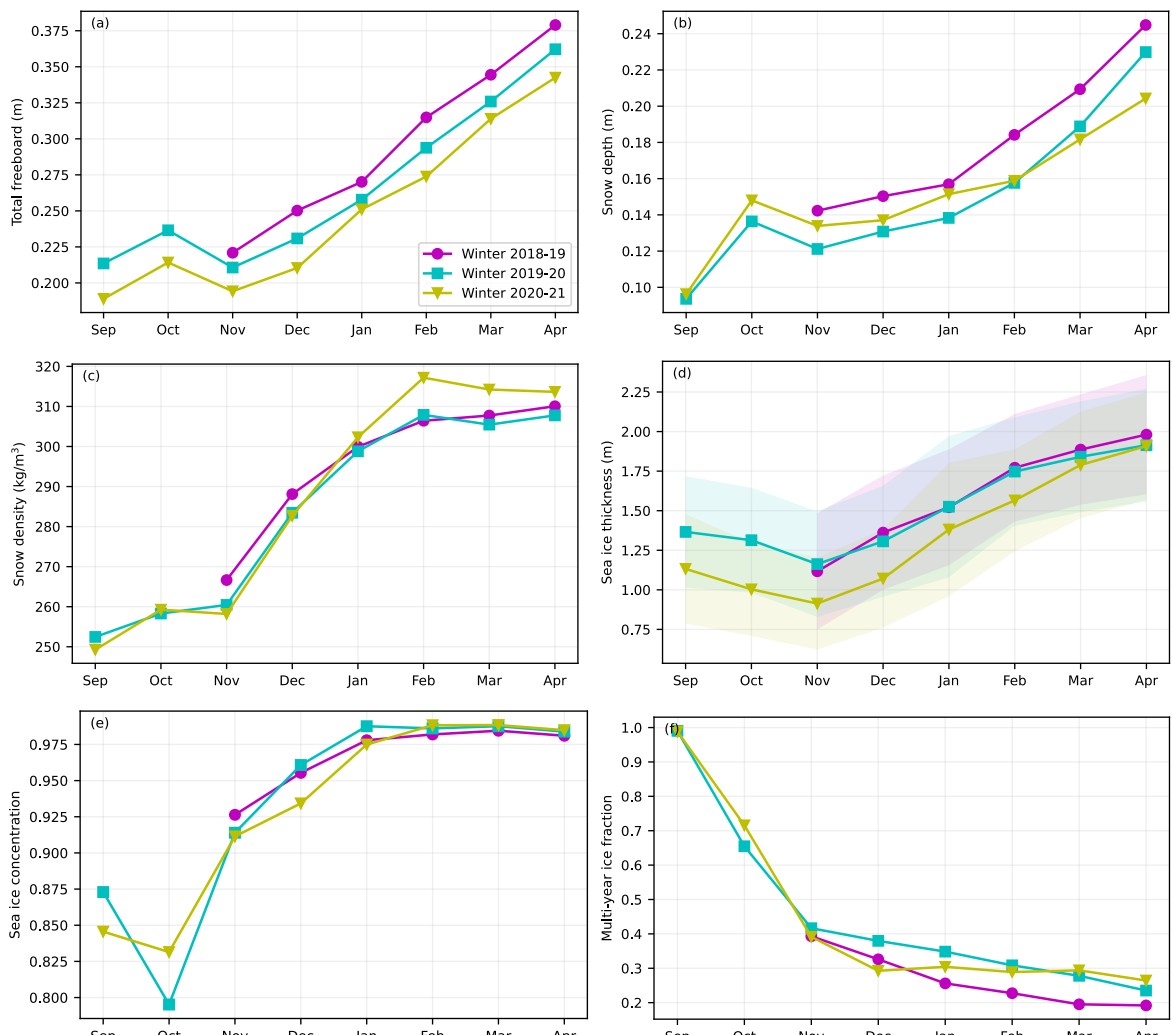

**Figure 12:** Time series of monthly mean ICESat-2 Inner Arctic Ocean sea ice freeboard (top left), redistributed NESOSIM v1.1 snow depth (top right), NESOSIM v1.1 snow density (middle left), OSI SAF multi-year ice fraction (middle right), CDR sea ice concentration (bottom left) and resultant sea ice thickness (bottom right) for the 2018/2019, 2019/2020 2020/2021 winters. Monthly means are generated using monthly gridded ICESat-2 thickness estimates (IS2SITMOGR4 v2, shown in Figure 8), within our Inner Arctic Ocean domain (Figure 5). The shading in the lower right panel represents the mean systematic thickness uncertainty.

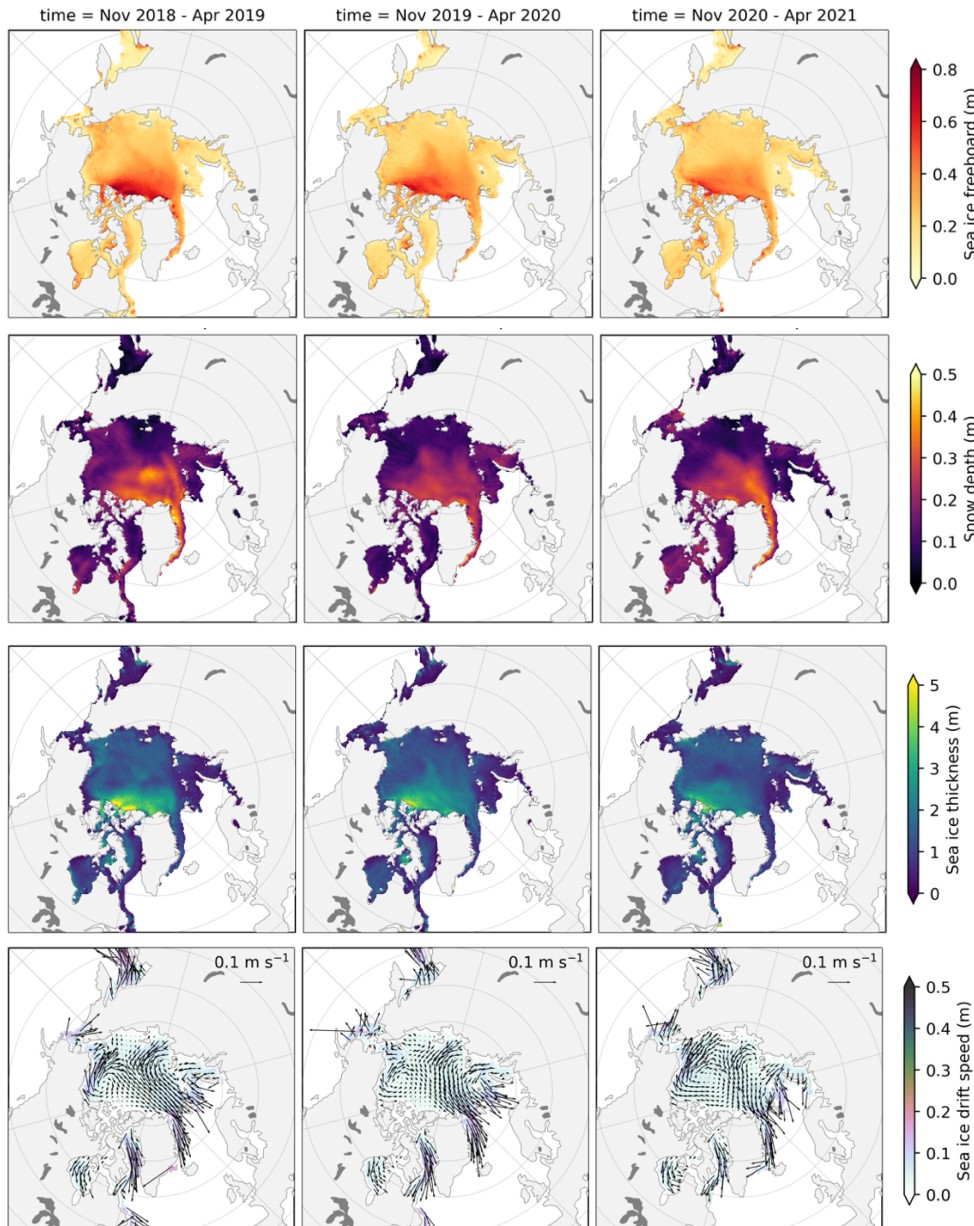

**Figure 13:** Winter (November to April) mean ICESat-2 freeboard (top) redistributed NESOSIM v1.1 snow depth (second row), sea ice thickness (third row) and OSI SAF sea ice drifts (bottom row) for the 2018-2019 (left column) 2019-2020 (middle column) and 2020-2021 (right column) winters based on the monthly gridded IS2SITMOGR4 v2 data (using the interpolated/smoothed variables for each variable). The thickness data are overlaid with winter mean OSI SAF drift vectors.

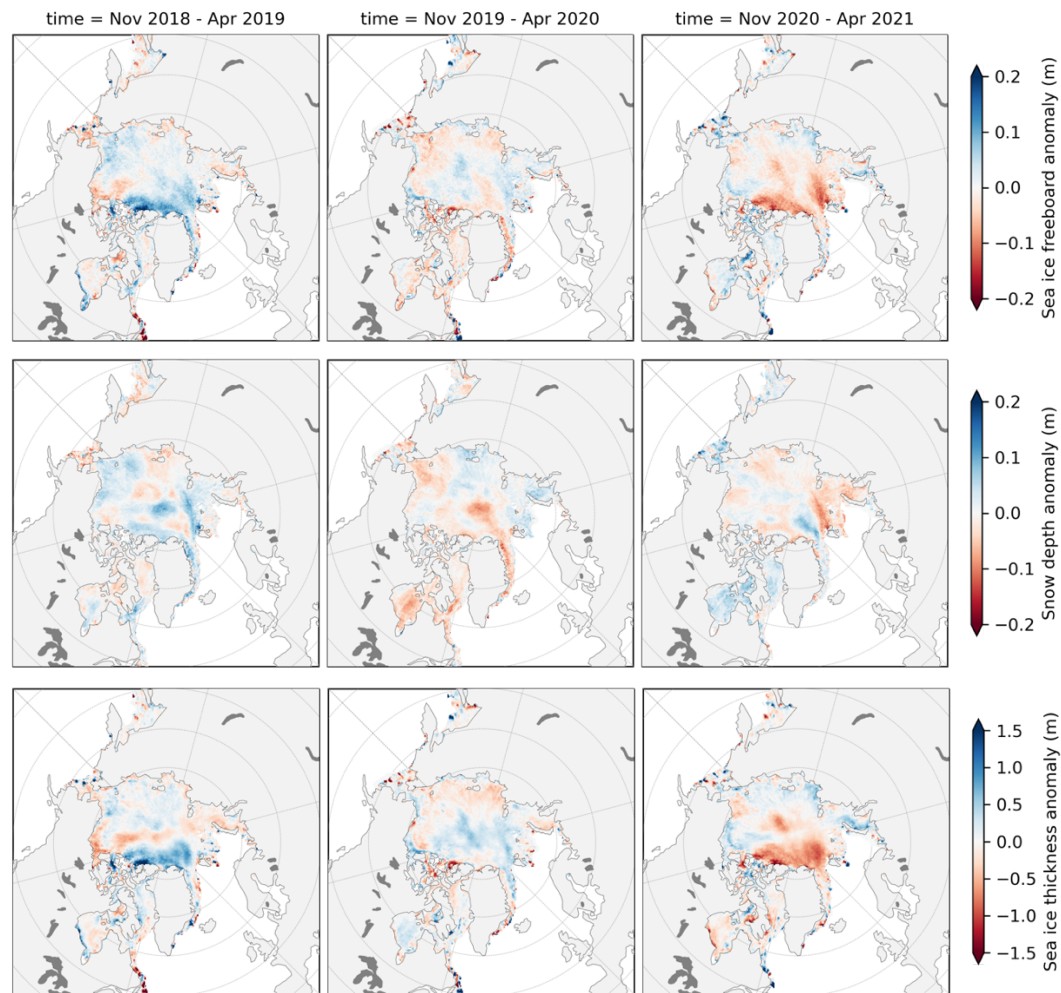

**Figure 14:** As in Figure 13 but showing the anomalies relative to the 2018-2021 winter means for the 2018-2019 (left column) 2019-2020 (middle column) and 2020-2021 (right column) winters.

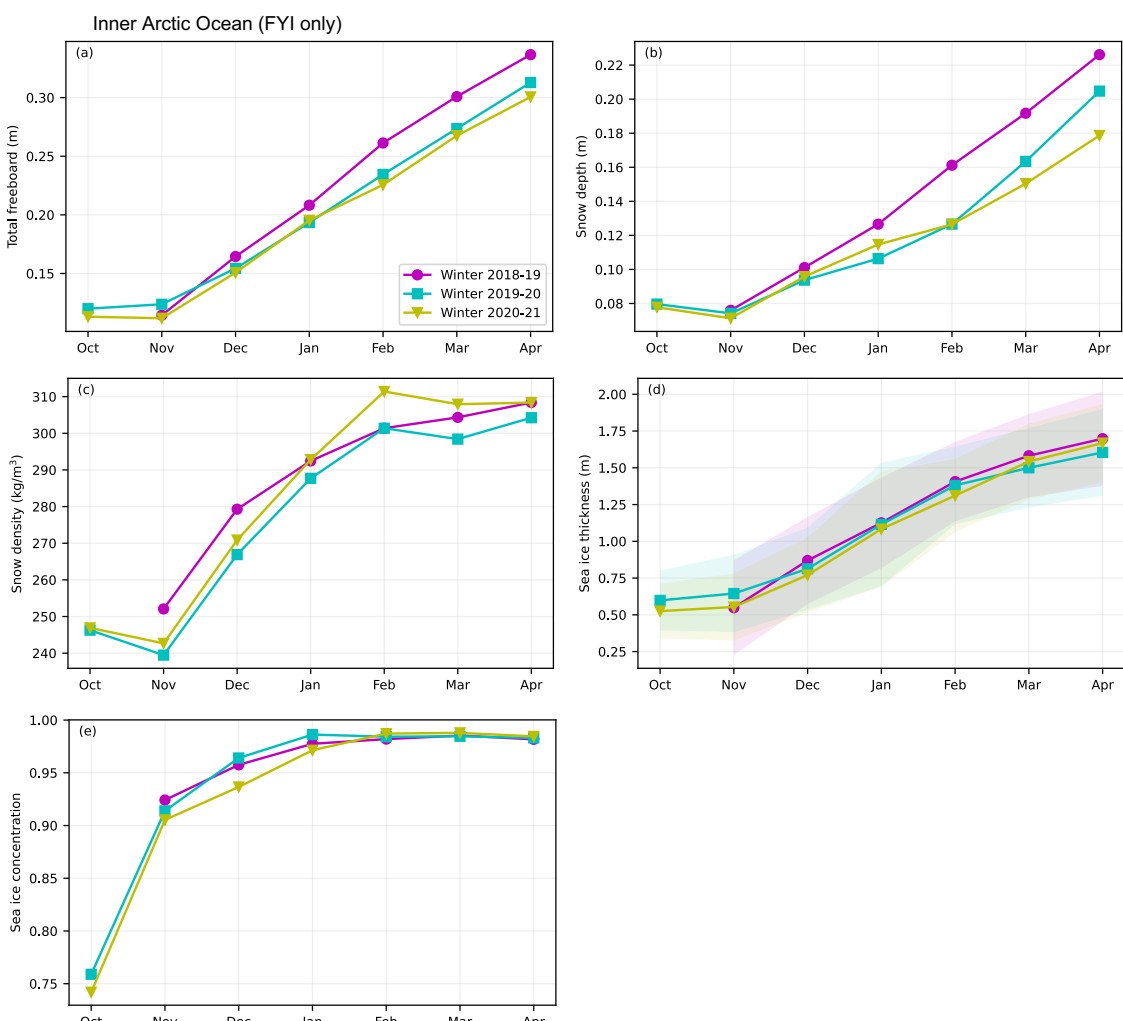

**Figure 15:** As in Figure 12 but for monthly means of grid-cells identified as first-year ice (FYI) only based on the OSI SAF ice type product. This figure also starts in October due to the lack of reliable FYI coverage information in September.

150

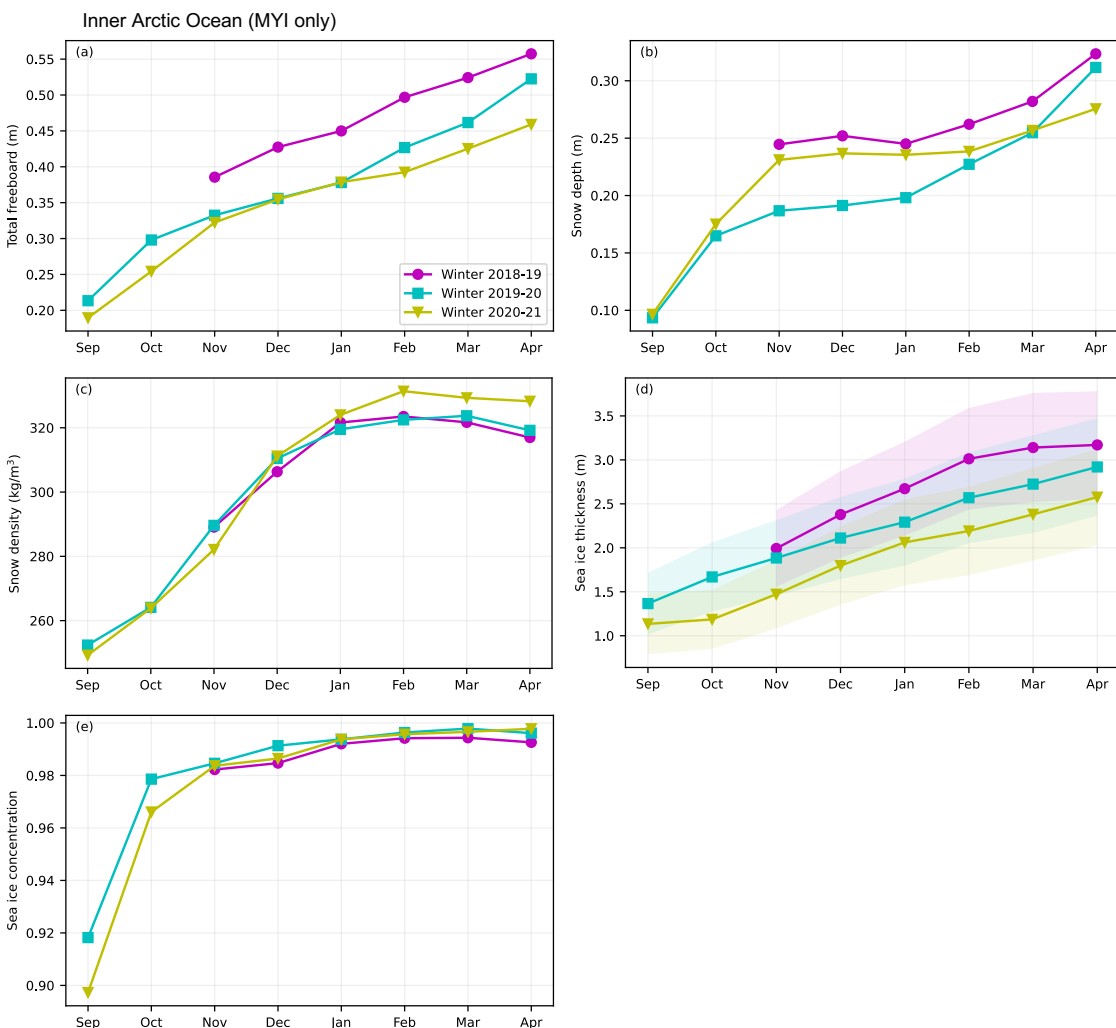

**Figure 16:** As in Figure 12 but for monthly means of grid-cells identified as multiyear ice (MYI) only based on the OSI SAF ice type product (no September ice type information is provided but we assume it is all MYI here).

155

