# Peer review of "Winter Arctic sea ice thickness from ICESat-2: upgrades to freeboard and snow loading estimates and an assessment of the first three winters of data collection"

_The Cryosphere, 2022_

## Author Response (AR1)

Response to Referee comment on "Winter Arctic sea ice thickness from ICESat-2: upgrades to freeboard and snow loading estimates and an assessment of the first three winters of data collection" by Alek Aaron Petty et al., The Cryosphere Discuss., https://doi.org/10.5194/tc-2022-39-RC1, 2022

Original referee comment in black, our responses in blue

**General comments**

The paper assesses the impacts of a number of changes to ICESat-2 ATL10 processing and to the NESOSIM snow model on estimates of along-track and gridded sea ice freeboard and ice thickness. This assessment is important for users of high level sea ice products such as ATL20 gridded sea ice. Overall the paper is well conceived and written. However, there are a nuber of issues that need to be addressed before the paper is ready for publications. I list these below. I also have a number of specific comments.

We sincerely appreciate the time taken by the referee to provide the review of our paper and the thoughtful suggestions. Please see below for our responses.

Overall, the quality of the figures is good. However, some of them could be improved by adding descriptive titles/labels to each panel. For example, figurure 7 has titles but these appear to be file variable names. Rather than "ice_thickness_unc", it would be more helpful to readers to have "Ice thickness uncertainty" spelled out. Likewise with panel (i) "ice thickness int" would be better as "Interpolated ice thickness". The authors might also want to think about a better layout and if all panels are necessary.

Yes, good points about the panel labels. We have improved and/or added these in the revised manuscript. We have decided to keep the panels as is other than edits/changes made based on other review comments. We have also removed the ice type panels from the time-series plots based on a later suggestion.

The Jupyter notebook is an excellent addition as is making the code available.

Thank you, we are excited about this aspect of our work!

Different releases are used for different evaluations. The authors show that there is little difference between releases 003 through 005 but it would make for a cleaner, and more up to date, analysis to use release 005 throughout. The only exception being to show differences between releases 002 and subsequent releases.

We have now updated the CryoSat-2 comparisons to use rel005 instead of rel003 data (the only place other than the release comparison where rel003 was used). Differences were negligible and we have updated the figures in the text accordingly. Based on a later comment we have also updated the CryoSat-2 comparison figure to improve readability. We still use rel004 in the snow comparison analysis however as that involved various different configurations that were all varied out during rel004 processing. Re-doing this would be very time-consuming and not add much value considering the negligible freeboard differences between rel004 and rel005.

I would like to see a map in the main paper showing the "Inner Arctic Ocean" region as the study region introduced as part of the methods. This would focus readers attention on the analysis region up front.

Agreed, we have now added this to the main manuscript as Figure 5.

Figure 8 is another example of a figure that would benefit from having labels such as a) sea ice freeboard. Parameter names are on the y-axes but they are small. Panels a, b, etc should be referenced in
the text.

We have now added these figure panel labels to Figure 8, and also to Figure 12 and 13. We did not feel it was necessary to provide further labels as these overly cluttered the figures when we tried this.

There are a number of places in the text where important statements are put in parentheses. I think it would improve readability to
rewrite these statements as part of the main text. Some of these parenthetical statements are unnecessary.

Agreed, we have removed several of these parentheses from the manuscript. It is a bad habit of the primary author!

**Specific comments**

L60. "is *being* developed"

Added, added.

L63. Suggest "collected to estimate sea ice thickness"

Added, changed.

Section 2. I think it would be helpful to summarise upgrades to IS2 processing, NESOSIM and ATL20 gridding in a simple table.

We didn't feel like a table was the best place to provide all this information. We currently include the Table (Table 1) summarizing the NESOSIM configuration upgrades, but IS2 processing upgrades were more descriptive so we think it's best to keep this to the description already included in the manuscript.

L111 prefer "km" to be consistent.

We followed the convention of the NSIDC user guides here and feel this is easiest to interpret for the user.

L124 "0-3 cm freeboard changes at basin scales". Does "basin-scales" refer to the Inner Arctic region used in the current paper? Maybe say "an increase in basin average freeboard of up to 3 cm.

Added, changed.

L139 Suggest "New releases of ATL07 and ATL10 also reflect upgrades to the underlying ATL03 processing, such as improvements in geolocation.

Added, changed.

L141 and 110. ATBD for ATL07/10 use "surface reference" rather than "reference sea surface". To avoid confusion it might be better to use the same terminology as the ATBD.

Added. We have used "sea surface reference" to be more consistent with the ATBD which sometimes uses this and sometimes shortens it to just sea surface.

Figure S1. Would it be better to have this figure in the main text? Also, the point here is that the number of reference surfaces is reduced from rel002 to rel003 because dark leads are not used. However, the count difference is positive. It make more sense to me to have this reduction as a negative number.

Agreed, we have moved this to the main manuscript and changed this to show counts relative to rel002. Now Figure 2.

[Figure]

Figure 2: (bottom row) number of 10 km along-track reference surfaces from the three strong beams from November 2018 to April 2019 for Release 002/rel002 (left) to Release 005/rel005 (right). Panels above show the difference in reference surface counts from releases rel003, rel004 and rel005 relative to rel002.

L190 Effectively the /beta and /gamma terms in equation 1 are corrections to solid precipitation. It is not clear to me what the difference is between the two terms. They could be combined into a single loss coefficient.

We introduced the atmosphere snow loss term as an added blowing snow to atmosphere loss term as we expect some fraction of snow that is disturbed during high winds will be

sublimated away to the atmosphere, which we developed to be an added function of the new unitless atmosphere loss parameter and not a function of ice concentration.

We agree that for simplicity it makes more sense to combine these coefficients in this equation which we have now done. We now refer to a "blowing snow atmosphere loss coefficient" which has units of $s^{-1}$ as in the now renamed blowing snow open water loss coefficient. The value has been reassigned by combining the two terms into one, now with a value of $2 \times 10^{-8}$.

L217 Do you mean "For each OIB snow depth product, snow depths are binned into 100 km grid cells using a drop-in-the-bucket averaging procedure. For each grid cell, the median snow depth of the three products is then assigned as the grid cell snow depth". So in all cases, you are taking the middle value. If the number of products was larger, I can see this as an acceptable approach to avoid outliers but for just three values, you can't really identify an outlier. It would seem that the mean is a better estimator.

Yes, that is the correct interpretation. We decided on using the median and not the mean as we are aware of some odd behavior in some regions and some dates for some of the algorithms. For example, Kwok et al., (2017) shows that the SRLD-derived snow depths exhibit a strong positive trend including very thin snow depths at the start of the OIB time-series, which appears unrealistic (as also noted by Kwok et al., 2017). We admit that three products is not much of a distribution to truly assess outliers, but we were keen for our results to not be consistently skewed by one algorithm being constantly biased compared to the others, hence our approach here.

L 230. "within reason" This needs some clarification. Are there limits you can set on depth or start date?

This is a concept we are currently exploring in the more sophisticated NESOSIM calibration efforts cited in the paper (Cabaj et al., 2021). Due to the lack of modern and reliable late summer/early fall snow depths at basin scales our knowledge is mainly heuristic: Arctic sea ice refreeze generally begins sometime in September following the Arctic sea ice extent minimum.

In response to this comment and a comment from reviewer #2 we have included more discussion on the SnowModel-LG end-of-summer snow depths, which generally show a complete removal of snow in August in those simulations, with snow depth increasing in September onwards.

"as another tuning parameter, constrained mainly by limited evidence in the literature. For example, the Warren et al., (1999) climatology (W99) shows a mean snow depth of 3 cm in August including depths of up to 8 cm near the Greenland/Canadian Arctic coastline based on the quadratic fit to observations. However, output from SnowModel-LG presented in Stroeve et al., (2020) shows zero snow depths in August in the earlier (1985/1986) and later (2015/2016) time periods of that time-series. As NESOSIM includes no snow melt terms, we prefer instead to initialize later in the year (Sep 1st) and prescribe an expected end of August mean snow depth based on our original temperature scaled W99 August climatology."

Figure 3. The left panel is busy. I suggest having a separate panel for October and April. The horizontal grid-lines should be lighter or removed.

Agreed, we have now split up the panels into October and April which has increased the readability of the figure.

[Figure]

Figure 4: (Mean Arctic snow depths in October (a) and April (b) from NESOSIM v1.0 and v1.1 within an Inner Arctic Ocean domain (Figure 5). NESOSIM now depths are also masked where concentration (from passive microwave) is less than 25%. The cross markers show the extended ICESat-2 NESOSIM v1.0 results used in (Petty et al., 2020). The dashed cyan horizontal lines show the NESOSIMv1.1_2010-2020ave snow depths averaged across the respective month, while the dashed black lines show the modified Warren climatology (mW99) in October and April respectively for regions of coincident NESOSIM v1.1 coverage. (c) violin plots showing interannual distributions of monthly mean snow depths from NESOSIM v1.1 within an Inner Arctic Ocean domain from 1980-2021, colored markers indicate mean monthly snow depths for recent (ICESat-2) years.

Note that there was an error in the original figures in how the Inner Arctic Ocean region mask was applied which = has been fixed for these new plots. The main difference is a small shift in all snow depths and a small relative increase in the April mW99 snow depths, which were lower in the original manuscript, and some slight differences to the violin plot ICESat-2 snow depths.

We have also added new maps to the SI (now Figure S1) showing maps of the differences between the NESOSIM 2010-2020ave and mW99 snow depths for October and April too – highlighting the regional differences including strong differences in the Kara Sea region especially.

[Figure]

**Figure S3:** Comparisons between NESOSIM v1.1_2010-2020ave and the modified Warren snow depth climatology (mW99) within our Inner Arctic Ocean domain. The mW99 snow depths are limited to valid NESOSIM grid-cells. Top row shows October (Oct) comparisons while the bottom row shows April (Apr).

L254 One of the arguments for not using the Warren climatology for snow depth is that it is not representative of the present day conditions. The previous paragraph and Figure 3 have been used to argue that recent years snow depth are also lower than average and may be declining. So why would you use a climatology of NESOSIM.

We were wrong to refer to this as a climatology. We now refer to this as a 'modern era representation' and have labelled this dataset NESOSIM_v1.1_2010-2020ave to be clear we only use data from the recent 10-year period to better reflect current conditions.

Wouldn't using output from an operation product or low latency reanalysis be a better option?

It is our understanding that there is a wide-spread in snow depths across available operational products which are generally not calibrated to available observations (e.g. Operation IceBridge). We agree there is potential there but are unaware of a reliable product for this purpose with consistent output.

L266. The redistribution method needs a reference.

Added the Petty et al., (2020) and original Kurtz et al., (2009) citation.

L296 The smoothing/gridding procedure needs more explanation. It would be helpful to say why each of the steps are done. Why use Delaunay triangulation - generally this method is used to interpolate unstructured data? Presumably the KDTree algorithm is to speed up the search for neighboring cells.

We have reproduced all maps and the time-series analysis with the non-interpolated data. This was previously a mix, however our testing has shown the difference in results to be negligible when considering basin-scale aggregates. Our main motivation for introducing the interpolation was to fill in the pole hole and increase spatial coverage and mitigate spatial biases when comparing across months/years, especially considering some of the declines in coverage since the removal of dark leads from the freeboard derivation. However, we admit our method here is crude and do not wish this to distract from the main analysis which is now entirely based on non-interpolated gridded data and we have removed and reordered the discussion of the gridded dataset and the interpolated variables.

L467: "We also include in this Version 2 IS2SITMOGR4 dataset smoothed and interpolated variables of freeboard, snow depth and thickness in an initial attempt to fill in the pole hole and mitigate the spatial sampling biases. These preliminary variables are not used in the subsequent analysis presented here but are available to interested users and shown in the Jupyter Book discussed below. We expect that future work will explore more sophisticated interpolation procedures and blending with other thickness datasets, which we discuss more in the summary section."

Some notes on the method: Delaunay triangulation interpolation benefits from being a flexible interpolation procedure for our needs. The input data is already gridded but the interpolation is able to fill in gaps across variable sizes and directions, e.g. gaps across vertices and more than one grid-cell away. Although we admit simpler algorithms could have been used, this is already a built-in function with the core scientific Python ecosystem making it easy to implement for our needs. Similarly, the KDTree algorithm is perhaps overkill as the problem being solved (finding distance of each grid-cell to the nearest raw valid grid-cell) is a relatively simple one considering the fact the data is already gridded and not too large. We've decided to drop the mention of KDTree as that isn't needed here. Note that the scripts used to generate the entire gridded datasets will be made available so interested readers will be able to reproduce our exact methodology if needed. The interpolation code is already contained in the Jupyter Book which we have now highlighted in the revised paper.

Figure 4, L343. How do these look for other months and for other years? No need to show them but a comment in the text would be helpful.

It was actually challenging to maintain all the needed datasets across different versions and releases so we only have these months on-hand to discuss. We feel they represent the key changes through the winter accumulation season. We look at monthly changes in the gridded comparisons which show no obvious step-change or anomalies in the results.

L354. Significant or major?

Agreed that is better language, made the change.

L356. Prefer peak rather than mode. Mode could be confused with operating mode.

Agreed that is better language, made the change.

Figure 4. How many segments are used to generate these plots? Are dark leads more common in November?

The issue of lead counts and dark vs specular in ATL07 was discussed more in Kwok et al., (2021) and Petty et al., (2021). PDFs of January, June and October 2019 SSH separated by leads counts are shown in Figure 3 of Kwok et al., (2021) showing specular lead counts dominating over dark lead counts. Dark lead counts increased through to January as overall coverage increases also.

In our thickness processing we do not store information regarding the surface type (e.g. specular or dark lead) as we're only using freeboard and associated variables, e.g. segment length, to generate our thickness estimates. We could add the freeboard segment counts to the plots if desired - it is a lot, but drops from rel002 onwards as the ref surf count plots highlight - but we do not believe this would add much value.

L370. The name NESOSIMv1.1clim has not been introduced yet.

Agreed. In Section 2.2.1. we have now referred to this as the modern-era mean and included the relevant label 'NESOSIM v1.1_2010-202ave' here and elsewhere in the manuscript and figures.

L389. Suggest "In Figure 6, we show the correlation coefficients, mean bias and standard deviations of ICESat-2 monthly gridded ice thickness from rel002 and rel003 compared with ESAs CryoSat-2."

Agreed, we have used this and also added a bit more clarity to this and the following sentence.

What are the standard deviations of? Why mask data less than 0.25 m?

Standard deviation of the differences, so the standard deviation between the two after removing the mean bias. We have clarified this based on the previous comment.

Also added "to focus more on the representation of consolidated pack ice between the two sensors, rather than the added complexities of thin and marginal ice."

Figure 6. I suggest removing the shading and, for each month, plot release 002 and release 003 as separate columns. That way you can see the ovelap. The shading suggests the data is continuous rather than discrete monthly data.

Agreed, we have now reproduced these plots as bar plots which we think has improved the readability. Based on the earlier comment these are now also base don rel005 ATL10s.

[Figure]

*Figure 8:* *Comparison statistics of monthly gridded CryoSat-2 thickness for four different CryoSat-2 products (GSFC, JPL, CPOM, AWI) with monthly gridded ICESat-2 sea ice thickness using rel005 ATL10 and the same snow loading and ice density input assumptions from November 2018 (11-18) to April 2019 (04-19). Data are compared within our Inner Arctic Ocean domain and for grid-cells in both datasets that contain thicknesses > 0.25 m.*

L445. NESOSIM presecribes snow density for new and old snow. The bulk density is a weighted average of these two values. How much can be read into variations in density?

This is a fair point, the parameterization is crude and we offered this up more to understand its impact on thickness. We have added the following:

"Due to the crude nature of the NESOSIM density parameterization, we do not view this analysis as a reliable interannual snow density assessment but highlight this more to understand the density variability impact on our ice thickness estimates."

Figure 8. Why is sea ice concentration lowest in October? Is this an artifact of averaging.

This is largely an artifact of coverage changes due to sea ice refreeze and the fact we don't include grid-cells with a mean concentration <50% in ATL10. We have added the following:

"Note that the concentration decline from September to October is due to changes in data coverage as regions with ice concentrations < 50% are not included in ATL10. "

Figure 9. The flow vectors obscure the thickness data. They are not really discussed. Are they necessary? Could they be relegated to supplemental material?

We have now added these overlaid on drift magnitude as a new row in the figure (now Figure 11).

[Figure]

Line 535. Care needs to be taken with ERA5 (or any reanalysis) near-surface variables over snow. ERA5 snow parameterisation is still a single layer, which does not produce realistic surface fluxes (Arduini et al 2019).

Agreed, we did not include surface turbulent fluxes partly for this reason.

L540. Are three years of data enough to make a statement about strength of coupling?

Our point here was that near-surface conditions are generally expected to be strongly coupled with the sea ice state. We did not mean to imply we are discovering this coupling by our analysis here. We have re-worded this line and other elements of this paragraph to make clearer what our limited analysis has shown.

L591. This seems to contradict what is shown in Figure 4.

We state height biases here which are different to the relative measurement of freeboard. We have added 'absolute' to make this clearer.

Figure 12 and 13. The multi-year ice fraction panel is not needed.

Agreed, we have dropped this and the FYI panel in the new figures.

Arduini, G., Balsamo, G., Dutra, E., Day, J. J., Sandu, I., Boussetta, S., & Haiden, T. (2019). Impact of a Multi-Layer Snow Scheme on Near-Surface Weather Forecasts. Journal of Advances in Modeling Earth Systems, 11(12), 4687–4710. https://doi.org/10.1029/2019MS

References

Kwok, R., Kurtz, N. T., Brucker, L., Ivanoff, A., Newman, T., Farrell, S. L., King, J., Howell, S., Webster, M. A., Paden, J., Leuschen, C., MacGregor, J. A., Richter-Menge, J., Harbeck, J., and Tschudi, M.: Intercomparison of snow depth retrievals over Arctic sea ice from radar data acquired by Operation IceBridge, The Cryosphere, 11, 2571–2593, https://doi.org/10.5194/tc-11-2571-2017, 2017.

Kwok, R., A. A. Petty, M. Bagnardi, N. T. Kurtz, G. F. Cunningham, A. Ivanoff (2021), Refining the sea surface identification approach for determining freeboard in the ICESat-2 sea ice products, The Cryosphere, 15, 821–833, doi:10.5194/tc-15-821-2021

Petty, A. A., M. Bagnardi, N. T. Kurtz, R. Tilling, S. Fons, T. Armitage, C. Horvat, R. Kwok (2021), Assessment of ICESat-2 sea ice surface classification with Sentinel-2 imagery: implications for freeboard and new estimates of lead and floe geometry Earth and Space Science, 8, e2020EA001491. doi:10.1029/2020EA001491.

Response to Referee comment on "Winter Arctic sea ice thickness from ICESat-2: upgrades to freeboard and snow loading estimates and an assessment of the first three winters of data collection" by Alek Aaron Petty et al., The Cryosphere Discuss., https://doi.org/10.5194/tc-2022-39-RC2, 2022

Original referee comment in black, our responses in blue

This paper discusses improvements in the ICESat-2 (IS2) processing of sea ice thickness retrievals from different releases of the IS2 products. As such it feels more like a NASA technical report that discusses how the different versions change the thickness retrievals. The author previous published in 2020 on the processing chain to IS2 and I do not find that with the changes this now warrants an updated assessment of thickness changes and a new publication. The question is what do we really gain from this paper vs. having a NASA technical report on the changes in data processing?

This is in part because any sea ice thickness (SIT) assessment depends strongly on the choice of snow loading used. It also depends strongly on the choice of snow depth processing applied to OIB data for validation of your snow loading, and the seasonality of this validation period. It seems that with the changes presented to NESOSIM there are minimal changes anyway to the snow loading and thus it is the changes to the lad detection that seem to have the largest influence. To make this paper more impactful and not just a technical report on updates to IS2 data processing, one way forward could be to assess the choice of snow loading in the IS2 SIT retrievals. Since Zhou et al. (2020) already showed how different these data products can be, and other studies such as Mallett et al. (2021) and Glissenaar et al. (2021) detailed how using different snow loading can lead to different trends, one really cannot trust any assessment of thickness changes over the 3 years evaluated here without addressing the uncertainty in the snow loading. How would Figures 8-10 look using different snow data sets for example? You state that it's the freeboard processing that results in the largest changes (again indicative that this should be a technical report), but given the wide variety of snow depth data sets out there, the 3 years analysed here may be quite different depending on data set applied. And is analysing 3 years of data really useful for assessing drivers of SIT variability? At the moment I really do not see much value in having this as a publication in The Cryopshere for an incremental update to the IS2 processing chain. That doesn't mean it shouldn't be published someplace, but The Cryosphere should be for more impactful papers.

We thank the reviewer for taking the time to provide this review. There are a few general issues raised here that we address first:

*Technical report:* we feel that a peer-reviewed paper in The Cryosphere is a highly suitable place for this work. The thickness data we present here is not an official mission product so has no such Algorithm Theoretical Basis Document (ATBD) or detailed technical reporting infrastructure in-place. The associated NSIDC user guides for both the along-track and gridded datasets shown here provide only top-level information regarding data production and notable changes as in other NSIDC products. All official ICESat-2 products including ATL07 and ATL10 are described in Algorithm Theoretical Basis Documents (ATBDs) which include change logs and descriptions of updates to the underlying data processing. However

even they do not include results of these changes on the data output or downstream impact assessments, hence the need for papers like Kwok et al., (2021, The Cryosphere) for highlighting the rationale and impacts of algorithm changes on basin-scale freeboard distribution.

A major aim of this paper is to highlight the changes to the ATL10 freeboard product across several releases (rel002 to rel005), together with updates to NESOSIM and, most importantly, the impacts of these changes on our estimates of winter Arctic sea ice thickness. As freeboard is largely measured by satellite altimeters like ICESat-2 towards the goal of inferring estimates of sea ice thickness, we believe a manuscript detailing these changes and their impacts is highly warranted and also scientifically insightful, especially considering the three years of data we now have and show from ICESat-2.

We do not plan to assess every new release of ATL10 in this manner, but considering this is the end of the ICESat-2 3-year prime mission period, assessments regarding data quality and impacts on higher-level products are urgently needed considering the importance of this mission and potential for improving our understanding of sea ice conditions.

*Differences in snow loading:* the primary author was involved in both Zhou (2020) and Glissenaar (2021) studies so is well aware of issues surrounding snow loading uncertainty and impacts on thickness. We use NESOSIM as it is configured to produce daily data across the entire Arctic Ocean including its peripheral seas and data is available for our entire study period. NESOSIM v1.1 was calibrated against a new consensus snow depth estimate from OIB giving us additional confidence regarding its reliability compared to other products available (some are calibrated, others are not). The Zhou (2020) study showed that NESOSIM output was largely consistent with other products but assessments of accuracy are still hindered by the lack of contemporary ground-truth data. We have added the following to the summary section:

"Recent studies leveraging newly generated Arctic snow reconstructions and satellite-derived data products, including the joint ICESat-2/CryoSat-2 derived snow depths, are helping collectively provide new insights into snow depth variability and its impacts on sea ice thickness and its contribution to total thickness uncertainty (Zhou et al., 2021; Mallett et al., 2021; Glissenaar et al., 2021). While these datasets, including NESOSIM, are still generally limited by a lack of contemporary ground-truth data for assessing data accuracy, the creation of new operational, i.e., continuously updated and disseminated, snow products should help enable more comprehensive assessments of systematic snow loading uncertainties."

Our derived thickness data includes an estimate of thickness uncertainty which includes a contribution from both random and systematic uncertainty from snow loading. We use NESOSIM together with the modified Warren climatology to deduce the latter which is represented in the shading in our thickness time series plots together with the other potential sources of systematic uncertainty. A similar paper using CryoSat-2 with ICESat-2 to infer snow and thickness concurrently for the three-year period of data we have was also published in GRL after this discussion period started (https://agupubs.onlinelibrary.wiley.com/doi/epdf/10.1029/2021GL097448). They show very similar thickness results to our NESOSIM-derived estimates which we now highlight in our revised manuscript (in the results and in the summary) as an indirect validation of our results. It is also worth noting that this analysis did not include any thickness uncertainty estimates.

It is our belief that the increased focus on snow on sea ice in recent years will help provide a more complete estimate of its contribution to thickness uncertainty, but more work is

needed to ensure timely and consistent data access as we now discuss more in the summary.

*Three years of data:* That is unfortunately all we have from ICESat-2, and we strongly feel that the results we show will be highly informative to a wide spectrum of readers interested in Arctic sea ice variability.

More specific major comments:

It is stated that NESOSIM is updated to use ERA5 calibrated against CloudSat and a new blowing snow term. However, there is no validation of this blowing snow loss term, or discussion on how the coefficients, i.e. wind action threshold, blowing snow loss coefficient and atmosphere snow loss coefficients are derived and validated. There is no in situ evidence that a significant amount of snow is lost to leads in the winter (any lead in winter quickly refreezes in a matter of a few hours), and there is no assessment here of the magnitude of this new snow loss term, and comparison to the old (and presumably still used) snow loss term to leads. Since SIT retrievals depend very strongly on the snow loading, at a minimum some quantitative analysis is needed on what these changes represent in terms of the overall snow mass, and some science justification is needed for doing this in the first place. It seems that some artificial tuning is based on trying to reduce the mean difference with OIB snow depths, but of course those are not perfect either. And they are done only in the springtime, and the question is how valid this bias- correction is for other months during the winter season?

We discuss the rationale behind NESOSIM development in the original manuscript (Petty et al., 2018) including the use of the snow loss terms as largely unconstrained free parameters. The maps included in Petty et al., (2018) show that the impact of the blowing snow lost to leads term, which we now refer to as "blowing snow open water loss" is isolated to regions of lower concentrations in the more peripheral Arctic seas where lead counts and widths are higher and large stretches of open water are prevalent, and where temperatures are warmer and winds can be stronger too.

We have added an additional comment about this in the revised manuscript:

L196: "As discussed in the original NESOSIM study (Petty et al., 2018), these snow loss terms are crude representations of complex physical processes that we introduce primarily to remove snow and improve correspondence with the limited observations we have for calibration purposes.

Based on review #1's comments we have also re-worded and simplified the discussion of the new blowing snow atmosphere loss term.

The author is wrong about what SM-LG does at the end of summer as it keeps the snow cover in places where it doesn't entirely melt out. Also, snow can start to accumulate before September in the Arctic, and thus it seems these changes are made purely to reduce your bias but there is no physical reason to justify these changes. I do not think that because NESOSIM matches mW99 in October that you can conclude you have "good" snow depths. In fact given delays in freeze-up, I would expect much thinner snow in October compared to mW99 based on the fact that ice is forming later than it used to.

We were incorrect when we stated that SnowModel explicitly removes all snow at the start of the simulation year (August 1st) as yes, theoretically, the model converts snow that is isothermal (0 °C) and saturated with meltwater at the end of a given simulation year to superimposed ice and enables the remaining snow to persist through to the following year. However, the related manuscript showing the output from SnowModel-LG (using ERA5 and MERRA-2 forcings) by Stroeve et al., (2020) shows zero snow depths across the entire Arctic in August and in some cases no snow in July either (Figure 2 of Stroeve et al., 2020). We have re-worded this line accordingly in the revised manuscript.

"For example, the Warren et al., (1999) climatology (W99) shows a mean snow depth of 3 cm in August including depths of up to 8 cm near the Greenland/Canadian Arctic coastline based on the quadratic fit to observations. However output from SnowModel-LG presented in Stroeve et al., (2020) shows zero snow depths in August in the earlier (1985/1986) and later (2015/2016) time periods of that model output."

We have provided an updated comparison of our modern-era NESOSIM mean output with modified Warren (Figure S3 in the SI) which shows thinner October snow in NESOSIM across much of the Central Inner Arctic, but thicker snow in the Kara Sea – a region where mW99 was largely produced through extrapolation of the observations collected in the more central Arctic through the quadratic fit. We have made a note of this in the revised manuscript.

Zhou et al. (2020) showed large differences between the various atmospheric reanalysis- based approaches to snow loading as well as the remote sensing-based retrievals, with the SM-LG (Liston et al. 2020) providing more spatial structure to the snow depth/density distributions, whereas products such as NESOSIM are artificially smoothed products. I see you get around this by taking your smoothed products and then adding some artifical spatial structure to match IS2 resolution, but why regrid to 100km in the first palce? Anyone who has spent time on sea ice knows the snow is very heterogeneous and thus the artificially smoothed 100km NESOSIM product seems unrealistic. Some justification for regridding the snow depth to 100km is needed and why you think this artificially smoothed data set is a good representation of snow over sea ice. Also, the impact of the redistribution then to 30m resolution is needed.

There is a significant spatial scale issue between the meter-scale information obtained from ICESat-2 freeboard altimetry measurements and basin-scale snow reconstructions, e.g. NESOSIM, which are largely based on satellite input data with resolutions of 10s of kms. This is a big challenge!
      To reconcile this scale gap, our approach has been to utilize high-resolution snow depth and freeboard measurements from Operation IceBridge obtained across the Arctic which, despite uncertainties, we believe is really our best means of bridging this scale gap using a redistribution/downscaling approach. NESOSIM thus provides our estimate of the seasonal/regional snow depth and density distribution, the redistribution scheme then helps us attempt to bridge the scale gap. The motivation behind the snow redistribution was discussed more in the original ICESat-2 thickness study (Petty et al., 2020).

This is an imperfect state of affairs but it seems that significantly finer resolution snow modelling will require much lower resolution input data and/or more comprehensive statistical distributions of snow properties that are validated against field data to capture the small-scale dynamic sea ice/snow processes. Advances like this are only now being incorporated into state-of-the-art sea ice models, e.g. CICE, but we do hope to explore this more in future work.

We are also not convinced that the 25 km-scale 'spatial structure' is related to improved accuracy and do not believe it should be used as a metric like this. The ice drift products are noisy at daily time-scales and this is a primary factor for us smoothing these data when used by NESOSIM, as was discussed in the original peer-reviewed NESOSIM v1.0 paper (Petty et al., 2018). Even lowering NESOSIM to 25 km would still require us to consider a redistribution/downscaling. Our aim with NESOSIM is thus to generate seasonal snow depth (and density) estimates constrained by the available basin-scale, but very limited, OIB observations.

Some assessment of the impact of using different ice motion products is also needed. It is not true that updated ice motion from NSIDC is not available, and the author could have contacted the data provider for updated ice motion fields. Since OSI SAF and NSIDC ice motion vectors to not agree, how does this influence your results? It is also unclear now how the Warren et al. climatology is used, are you assigning MYI snow depths on September 1 based on W99 and then accumulating snow? And finally, I'm not sure why so much smoothing is applied to both the snow and SIT retrievals, and some justification for this is needed. What does your SIT data product really give us if so much smoothing is applied? Snow and ice are highly spatially variable and thus is this a data product that is really useful to the community if it is artificially smoothed? Wanting "pretty" maps is not a reason to do this.

The original Petty et al., (2018) study undertook a comprehensive sensitivity study into the impact of differences in ice drift, using 4 different datasets (NSIDC, OSI SAF, CERSAT, KIMURA, Figure 11 and 12) concluding that at basin-scales this is of secondary impact to snow accumulation, but can have important regional impacts. It is challenging to discern a clearly optimal drift data product, so our choice in forcing is primarily driven by data availability. OSI SAF and NSIDC show reasonable agreement as shown in Petty et al., (2018) and our subsequent assessments of both products.

The reviewer's suggestion that we seek pre-release versions of the NSIDC drifts through independently contacting the providers seems problematic to us. The typical lag time for release of these products has, from our past experience, been about a year, so it is apparent that considerable time and effort needs to be taken in the processing and validation of these products prior to release. Furthermore, our philosophy of taking a more transparent and open-science approach (using whichever datasets are fully publicly available, carrying out

reproducible and verifiable analysis through Jupyter notebooks, etc.) precludes this kind of exclusive approach to obtaining data.

I do not find much value in the CS2 to IS2 comparison. In particular, now suddenly the mW99 climatology is applied after spending much time discussing updates to NESOSIM. This seems to be only because you want to use existing products out there, which we already know are not realistic because they do not have a realistic snow loading representations. Instead, maybe comparison of the freeboards would be a better thing to do, as you can convert the IS2 snow freeboards to ice freeboards with your snow loading from NESOSIM. Then we can better understand differences on the ice freeboard level, and may be get some insights into where the dominant scattering surface from CS2 is located as well as the influence of surface roughness on the freeboard retrievals. The use of PIOMAS is also not useful in my opinion, it's a model and has known biases, so adding it here just distracts from the overall paper.
The abstract is too long and reads more like a technical report.

We do not believe this was a sudden jump, we explain the motivation and approach in detail in Section 2.4 as making sure we use consistent input data is crucial when carrying out these thickness comparisons, and the effort involved was not trivial. It was also the same approach as that taken in our original ICESat-2 thickness paper (Petty et al., 2020).

This was not a paper investigating CryoSat-2 scattering issues but products of basin-scale sea ice thickness, hence the focus on that higher-level data variable instead of freeboard. We also believe the thickness biases are also more intuitive to understand for the interested reader. Based on the comments of reviewer #1 we have now updated this to use rel005 data to be consistent with the rest of our analysis (differences are negligible) and have changed the comparison figure to bar plots to improve readability.

PIOMAS is well-used by the community so simply highlighting the seasonal and regional differences we believe to be a useful exercise. It is a model, but it is constrained by observations (mainly SST) and has been well-tuned over the years to provide useful thickness estimates used across various recent studies for assessing climate-scale variability and more regional sea ice changes. We have now adapted Section 2.5 to state this more clearly and provide further citations:

"PIOMAS data is commonly used in the sea ice community for assessments of Arctic sea ice thickness variability at regional and basin-scales (Tilling et al., 2015; Labe et al., 2018; Petty et al., 2018b; Schweiger et al., 2021; Moore et al., 2018)."

We have made some small edits to the abstract to improve readability.

**References**

Kwok, R., A. A. Petty, M. Bagnardi, N. T. Kurtz, G. F. Cunningham, A. Ivanoff (2021), Refining the sea surface identification approach for determining freeboard in the ICESat-2 sea ice products, The Cryosphere, 15, 821–833, doi:10.5194/tc-15-821-2021

Petty, A. A., M. Webster, L. N. Boisvert, T. Markus (2018), The NASA Eulerian Snow on Sea Ice Model (NESOSIM) v1.0: Initial model development and analysis, Geosci. Model Dev., doi: 10.5194/gmd-11-4577-2018.

Petty, A. A., N. T. Kurtz, R. Kwok, T. Markus, T. A. Neumann (2020), Winter Arctic sea ice thickness from ICESat-2 freeboards, Journal of Geophysical Research: Oceans, 125, e2019JC015764. doi:10.1029/2019JC015764

Stroeve, J., Liston, G. E., Buzzard, S., Zhou, L., Mallett, R., Barrett, A., Tschudi, M., Tsamados, M., Itkin, P., and Stewart, J. S.: A Lagrangian Snow Evolution System for Sea Ice Applications (SnowModel-LG): Part II—Analyses, 125, e2019JC015900, https://doi.org/10.1029/2019JC015900, 2020.

---

## Author Response (AR2)

**Review of Winter Arctic sea ice thickness from ICESat-2: upgrades to freeboard and snow loading estimates and an assessment of the first three winters of data collection by Petty, A. A. (review #3)**

Review responses to reviewer #3 (technically a second review round). Review in black our responses in blue.

Summary: With the ICESat-2 satellite laser altimeter launched in September 2018 the sea ice community has access to very high resolution observations of the sea ice topography and hence novel means to retrieve the sea ice thickness distribution. This manuscript details the impact of recent changes in the processing of the ICESat-2 data relevant for such retrieval. It details further the impact of updates of an important data set of ancillary information required for this retrieval: the snow loading. The manuscript convinces with comparably clear messages of these impacts mentioned and comes up with a presentation, interpretation and discussion of the seasonal development of the freeboard, snow and sea ice thickness values at basin scale.

The present manuscript falls into the category of work typically connected to new satellite missions when the (raw) processing is still ramping up and updates in the derived products are issued at a comparably high frequency. It is therefore of a quite technical nature - albeit it has a considerable fraction of geophysical interpretation. I was wondering therefore, whether the authors have considered to hand in this manuscript also into the journal "Earth System Science Data"? I'd say it is at the verge between both journals.

I should note that I read this manuscript in the second round of reviews - which explains why I refer to comments of the previous review at a few occasions.

Thank you for taking the considerable time to provide this detailed review, your effort is much appreciated, as well as the positive comments about our paper above.

Re: journal selection - we felt that presenting comparisons across the first three winters - including reporting on the significant thickness decline in the 2020-2021 winter - made it much more suitable to The Cryosphere than a more data description journal. In response to this review we have also added new comparisons with BGEP upward looking sonar draft measurements (only recently made available) and additional CryoSat-2 comparisons which have added considerably to the paper and we think adds even more to the suitability of this study for The Cryosphere. We describe these major upgrades here:

**1. **BGEP ULS comparisons:**

These were not added in direct response to a review comment, but do help respond to some of the concerns about the value of the CryoSat-2 comparisons and their utility for validating our thickness estimates. As stated in response to the direct comments, we never intended to use CS-2 as a validation of our data, it was instead an intercomparison to other commonly used thickness data. The BGEP ULS data are more commonly used for validation purposes, however, as we do here also. Note that the data were only made available earlier this summer and we recently completed the analysis we believe is very suitable for inclusion here for providing crucial early validation of our ICESat-2 ice thickness data which is lacking from the literature to-date (due mainly to the lack of available data and the early lifetime of the mission). The comparisons are very strong and we think add considerable weight to our study, see Figure 10 below. New data description, results and discussion have been added to the revised manuscript to highlight these results.

**Figure 10**: Comparisons of IS2SITMOGR4, v002 (Nov 2018-April 2019, September 2019-April 2020, September 2020-April 2021) converted to ice draft against ice draft measurements obtained by Beaufort Gyre Exploration Project (BGEP) upward looking sonar moorings. The mean of all IS2SITMOGR4 data within 100 km of the given mooring are used in this comparison.

**2. Updated CryoSat-2 comparisons:**

In response to comments here (and partly in response to the original reviewers) we have decided to change the CryoSat-2 comparison approach. This was also motivated by the fact we now have BGEP ULS data that provides a robust assessment of our IS-2 thickness data. We now use CS-2 in two ways:

- 1. We provide the same CS-2 'regional/monthly' comparison as before, but we substantially reduce the description of this analysis as recommended. Essentially, we show the figure and highlight the in all three stats the 'agreement' with CS-2 has improved since our version 1 thickness data we think this is still a good sign and important to communicate. We also still think there is value in showing this comparison using the same input assumptions to remove the impact of different snow assumptions and to get a sense of the monthly/regional biases between the two retrieval approaches.
- 2. We now provide a new figure showing various comparisons (one example shown below) between CryoSat-2 and our monthly gridded ICESat-2 data but for the mean Inner Arctic Ocean thickness timeseries. In essence this is a comparison of the various products' best guess at the state of Arctic winter sea ice thickness and for this we use our final thickness product too (NESOSIM v1.1 snow loading). This comparison builds on (and integrates) the PIOMAS thickness comparison we originally included and is intended to provide a sense of seasonal agreement between these various thickness products. This also includes upgraded AWI/SMOS thickness data (Ricker et al., 2017), a new all-season CryoSat-2 thickness dataset (Landy et al., 2022) and merged ICESat-2/CryoSat-2 thickness data (Kwok and Kacimi, 2022). We think this analysis, albeit basic, is useful for the community in providing a broad sense of how our thickness data compares to other well-used products. Again, more regional information is provided in the maps that are on the online Jupyter Book for more engaged users.

**More discussion of the CS-2 additions are included below in response to the specific review comments.**

**General comments:**

GC1: The authors need - throughout the manuscript - clarify and correct the usage of "total (sea ice plus snow) freeboard" and "sea ice freeboard". Currently, these terms appear to be mixed and they are in part misleading the discussion of the results. A laser altimeter observes the total freeboard. A radar altimeter is supposed to observe the sea ice freeboard. This must be corrected in the manuscript both in the text as well as in all figures and tables where this applies. You will find repeated notion of this in my specific comments. This is the main reason why I suggest that this manuscript requires another round of major revisions because it might require some time to correct this in an appropriate way.

**'sea ice freeboard' is the language used in the ATL10/NSIDC data descriptions:**

https://nsidc.org/data/atl10/versions/5 so we had followed that convention. Our general view is that when talking to a broad audience it is helpful to include 'sea ice' to clearly indicate what the geophysical quantity represents, as total freeboard may not mean much to non-specialists. However, it is true that we include discussion of laser and radar profiling in this paper and agree it can be helpful in these more specific instances to use 'total freeboard' to refer to the freeboard including the snow then 'ice freeboard' to refer to the freeboard without the snow, similar to what you suggest.

In the introduction we now introduce the idea of total freeboard. We have also edited the rest of the manuscript to refer to just 'freeboard' when we are talking about ATL10 measured freeboards, as repeated use of total seemed superfluous.

**We have also added the following clarification to the data section:**

"The laser returns are expected to track the snow-covered ice surface, so ATL10 is expected to provide a measure of 'total' (ice plus snow) freeboard. "

GC2: Despite a high fraction of technical details there are a few technical and/or methodological issues that, to my opinion, are not laid out sufficiently well. To these belong i) the calibration of the parameter gamma, ii) the evaluation of the NESOSIMv1.1 product for other months than April, and iii) the description of how you derived the sea-ice thickness uncertainties.

We tried to strike a balance in this paper between providing a clear documentation of changes to NESOSIM and the sea ice thickness datasets, without getting too technical. We address these comments in detail where they come up in the specific comments below.

Specific comments:

L19-21: Is an enhanced agreement with CS-2 based sea-ice thickness data that desirable given the fact that penetration depth of the radar signal into snow plus ice-snow interface processes and properties play a much larger role therein?

We don't use CS-2 as a validation dataset, but we believe increased correspondence between the two satellite-derived products is generally encouraging and worth noting, especially as our results strongly suggest this occurred primarily due to improvements in the ATL10 freeboard determination. CS-2 has challenges with possible biases from ice-snow interface processes, but the data have also been well validated against available airborne/in-situ datasets to produce what we see as a reliable indicator of seasonal sea ice thickness change across the Arctic.

In response to a further comment below, we have now added additional CS-2 comparisons, including those from newer state-of-the-art CS-2 products to those originally presented in the manuscript as discussed above.

L35-46: In this paragraph you speak of "sea ice freeboard" in the context of laser altimetry. I strongly recommend to clearly differentiate between sea ice freeboard and total (sea ice plus snow) freeboard which is the quantity derived from the ICESat-2 elevation measurements first. Any conversion into sea ice freeboard (if need be) requires knowledge of snow thickness and density. It could be the that product you are using is named "sea ice freeboard" but physically it is not.

Yes, see main comment response above.

- What I am missing in front of this paragraph is that classical paragraph that tells us why satellite altimeters are such an important tool to observe the polar regions and why this is important. This would move your paper away from the impression one of the reviewers had that this is merely a technical report. In other words: So far the paper is not put into a larger context sufficiently well.

Agreed, we have now added an additional introductory paragraph about the importance of sea ice in the climate system and the need for new estimates of sea ice thickness.

L51-53: "Additional .... procedure" --> You could stress perhaps, that this is a high skill for daylight, clearsky conditions.

Yes, good point, we have added this now.

L108: "Sea ice freeboard" --> I assume this is in fact the total (sea ice plus snow) freeboard and should be referenced like this in the paper.

**Yes, see general comment response above.**

Section 2.1.1: I can understand that one of the previous reviewers got the impression to read a technical report rather than a scientific paper. This section appears to be quite long given the comparably little information that appears to be relevant for the content of the paper. To my opinion, this section could be condensed such that the main changes that determine the differences between rel002 and rel005 are highlighted while the reader is referred to the (regularly updated) technical documentation and change log associated with every release for the less relevant changes.

One of the main purposes of this paper was to highlight the impact of ATL10 changes on our estimates of sea ice thickness, hence the inclusion of what we see as very brief summaries of these pertinent

changes. We made this as condensed as possible, and already include a link to the ATBD change log where the more minor issues or issues not relevant to freeboard are described, so we don't think there is much more we can really do here.

L195: How is the calibration of gamma carried out?

This is discussed further down in this section "We heuristically calibrated NESOSIM v1.1 using the daily OIB consensus gridded snow depths with the aim of removing the mean bias relative to OIB when using the default NESOSIM v1.0 parameter settings ... tuning the new atmosphere snow loss coefficient,  $\gamma$ ,"

L212/213: "Here we choose instead ..." --> This recalibration has the disadvantage that is is only valid for April snow conditions. Is that correct? How reliable can NESOSIM v1.1 then be for the rest of the freezing season. How is this period evaluated?

Unfortunately, this is a consistent challenge with simulating snow on sea ice -a real lack of regionalscale observations beyond what is offered by Operation IceBridge which are mainly at the end of the accumulation season.

We feel the following text included in the manuscript, which we have also added 'early-season' to make the point clearer, is about all we can say:

"In the absence of contemporary early-season ground-truth data, we view the initial conditions (either their distribution or the representative start date) as another tuning parameter"

L232-241: It is still not clear to me after these lines what the initial snow thickness on the sea ice on September 1 is.

Some example maps for 2012/2013 are given in Petty et al., (2018). We have edited the test her to point the reader to that:

"based on the temperature scaled W99 August climatology shown in Petty et al., (2018, see Figure 2)."

L232: It is still not clear how you tuned gamma.

See response to the comment above on heuristic/manual calibration to the median OIB snow depths.

L251: "declining trend" --> Is suggest to either write "decline" or "decrease" or "negative trend". A declining trend is a trend which is not stable over time but where the change of the parameter with time is decreasing over time.

**Good point, changed.**

- I am sure you are aware that this decline in snow thickness has several causes, beginning with the change in ice age, over changes in snow accumulation (period), actual ice drift vectors and retrieved ice drift vectors, to potential inconsistencies and spurious trends in the input data to NESOSOM v1.1. Because of these it might be a good idea to not overinterpret this decline. In addition: The tuning / evaluation of the NESOSIM v1.1 is limited to April ... and hence the time series shown for October could be less credible / reliable than the time series shown for April.

Yes, good points, agreed, we have further reduced this trend discussion in the revised manuscript.

L272-276 / Figure S3: Please check these lines; it seems they contain two times almost the same sentence.

**Yes thanks, a similar line was erroneously added in the review responses. Removed now.**

- How much ice is in the Kara Sea in October? I am not overly convinced that this particular region is well suited looking at October snow thickness on sea ice. I am wondering where the ice edge drawn in the Kara Sea for October comes from.

Yes, good point, ice concentration is very low in the Kara Sea in October and NESOSIM (and Warren) are highly uncertain this region. We have now masked this plot using an 80% threshold (needs to be high as NESOSIM only provides data where ice concentration > 15%) – now virtually all the grid-cells in October in the Kara Sea have gone.

However, following other comments and feedback we have now dropped the climatology analysis from the manuscript.

We have included a new figure showing the mean NESOSIM v1.1 vs mW99 for the years analysed here , which we include instead

- Figure S3 units are cms. Is this correct?

This has now been removed from the manuscript as we removed the NESOSIM clim discussion entirely as it distracted from our other messages. We have instead included a new figure showing mW99 vs NESOSIM for the years analysed here (2018-2021) and have fixed the units accordingly (new Figure S4)!

- The color table used in the snow thickness maps on the left does not convince me that there is "measurable" snow thickness on Kara Sea sea ice. It looks white.

**See above comments.**

- Apart from these comments, I guess I have a conceptual problem with on the one hand striving to produce a temporally reasonably fine resolved snow thickness data set for proper sea-ice thickness retrieval and on the other hand generating (again) kind of a climatology.

Shouldn't the strategy be to use the auxiliary data sets when these are fully available? I mean, you also switched to a different daily sea-ice motion product for the time period that NSIDC drift is not available. You are not computing a climatology. Also, since you don't show these NRT products in your paper and only deal with the winter season 2020-2021 as the most recent, I believe you could condense this part of the investigation considerably. Perhaps you could mention at the side that a climatology based on NESOSIMv1.1 based on the 2010s (which would exclude 2020 by the way) provides an October snow thickness which is about 1 cm smaller than mW99 and an April snow thickness which is about 5 cm larger than mW99 for the inner-Arctic region you selected. You could then argue that most likely - if one still wants to use a climatology - then the NESOSIMv1.1 climatology might be a better choice ... but your results in fact only show the difference but do not indicate which of the two data sets is the more accurate one. I therefore find this comparison between mW99 and the 2010s snow thickness climatology not overly useful and I cannot recommend to keep this in the manuscript. To my opinion this sets the wrong signal.

Due to the inclusion of the new analysis and the issues you raise about the use of the NESOSIM average here, we've decided to drop this analysis from the paper.

- Did you try to compute a similar time series as shown in Figure 4 using W99 / mW99 and available multiyear ice areal fractions for "your" Inner Arctic region to see whether the negative trend in NESOSIM snow thickness is also visible in W99 snow thickness (simply via the change of the multiyear ice versus first-year ice partition)?

This was a good idea, although we're limited by the fact OSI SAF ice types are only available back to 2009. We have now calculated this using the interannual/monthly ice type and we think it is nice to show this to highlight the ice type dependency on the mW99 results and how that compares to NESOSIM. The results are quite intuitive, the variability appears similar in October, when snow is thinner and largely ice type dependent, but NESOSIM appears to have higher variability later in the season. We have made some comments on this in the revised manuscript.

L279-298: Given the coarseness of the NESOSIM snow thickness data I am wondering whether from the view point of spatial scales involved it would make much more sense to discard the interpolation from 100 km x 100 km over more than 4 orders of magnitude to 30 meters and instead work with 10 km along track mean freeboard estimates. With that the interpolation would just be about one order of magnitude - at least into one direction.

But lets see what you will write about this in your discussion section.

We do not really interpolate the 100 km data to the segment scale, what we explored in Petty et al., (2020) is fitting a distribution of segment-scale data within 100 km aggregates. As explored in Glissenaar et al., (2020, see references below) there is no real skill in this redistribution when using CS-2 like footprints (scales of kilometers), we think the skill really only starts to be shown when you redistribute to much higher resolutions, e.g. IS-2, and the difference between low and high snow depths over different types of ice surface becomes clear in the distributions.

- I note that your description of the error estimation in Lines 284/285 is not overly specific. If there are no further details given elsewhere I recommend to specify better how you carried out this step.

This information is provided in the linked to Petty et al., (2020) reference, so we do not believe it is worth repeating this here.

L307: So in the monthly sea ice thickness data you use the CDR version 4 but in NESOSIMv1.1 you use version 3. This difference in versions is possibly not dramatic, isn't it?

Yes, in our conversations with the CDR sea ice data producers it did not seem like there was a big difference. We switched the thickness processing mainly just to ensure our system is up-to-date for future processing needs.

L326-332: I am sorry, I don't get why you prefer to compare ICESat-2 sea ice thickness data based on rel005 of ATL10 with the four CS-2 sea ice thickness products using THEIR snow loading. What is the motivation? What are you aiming to show here? It seems you are using the CS-2 sea-ice thickness products as a benchmark against which you would like to reference your product. Is this a viable approach given the difficulties / assumptions these radar altimeter products need to deal with? If you find a large bias / RMSD ... I'd say this is fine ...

I think your last point gets at the approach - we did observe large biases between CryoSat-2 and our ICESat-2 thickness results in Petty et al., (2020) which were concerning and part of our rationale for looking into this again with our new thickness data. We think it's important to document clearly that the changes in freeboard have significantly reduced the large thickness biases. We agree that we do not intend for CryoSat-2 to be used as a benchmark to assess the ICESat-2 data, so following another comment have reduced the description of the comparisons shown in Figure 8 and focus more on the fact the biases have been substantially reduced.

As discussed earlier, we have now also introduced new CryoSat-2 comparisons, and crucially comparisons against BGEP Upward Looking Sonar draft measurements that are much more suitable to validate our thickness estimates (see major comment at the start of this response).

In addition, you only use data of strong beam #1 here - to be consistent with your previous work. Are you not interested in how your updated sea-ice thickness product (see the many updates you wrote about so far in the manuscript!) compares to all these other products?

Yes, based on various comments in this review and the release of new CryoSat-2 sea ice products (e.g. Landy et al., 2022), we have now updated our approach here to 1. Briefly show the improvements from Petty et al., (2020), before comparisons with ULS draft data and also seasonal CryoSat-2/PIOMAS thickness comparisons. In the latter we focus more on comparisons of 'best guess Inner Arctic Ocean sea ice thickness' on basin-scales and use the updated (v2) ISTSITMOGR4 data which does use all three beams. Please see description of this change at the top of the review responses.

Section 2.6: This section appears to need some more work at it seems not to be complete. Also, it is not clear what the ERA5 data are for here. "To assess the winter Arctic atmospheric conditions" reads as if you made an investigation of the atmospheric conditions such as comparing ERA5 with rawinsonde and in-situ observations or the like. You need to work on your wording. Also, the last sentence does not fit quite well. I guess you merely perform a consistency check whether ERA5 2-m air temperatures support your assumptions about the physics and conditions but the nature of what you do is far away from an "assessment". It is an inter-comparison.

We agree the analysis presented here is brief and our aim was to really provide some extra insight into whether the thickness changes (seasonally and interannually) are consistent with the primary atmospheric forcing differences. This use of 'assessment' seems to align with the definition of 'the evaluation or estimation of the nature, quality, or ability of someone or something.' Although again we acknowledge this effort is far from exhaustive. I am not sure inter-comparison is correct here (that would imply a comparison across different products perhaps?) so instead we have stated our aim is to "understand the possible relationships between seasonal/interannual differences in ice thickness and winter Arctic atmospheric conditions" and have worked on further clarifying our goals and the main results of this analysis.

L362-365: The derivation of individual freeboard values from surface heights and approximated sea surface level implies that there are negative freeboard values as well. What is their fraction and what did you do with these?

In the ATL10 algorithm, negative freeboards are set to zero. We have clarified this in the data section: *"Negative freeboards are set to zero."*

L366: The results shown in Figure 6 are for your inner Arctic region or the entire Arctic?

Thanks, results are shown in an Inner Arctic domain. We have clarified this in the figure caption.

L399: I see the "performance" of the v1.1\_2010s snow thickness climatology in a slightly different way than you. I would state that in April it appears to be a good representation of what Nv1.0 and Nv1.1 provide. But in January and particularly in November the distributions differ considerably, casting doubts about the credibility of this climatology; these doubts are justified looking at the sea-ice thickness distributions where usage of the climatology provides a substantially smaller modal (1st mode) sea ice thickness. If used for model initialization, assimilation or intercomparison studies the sea ice thickness data based on the climatology are certainly more problematic - especially at this critical time of the freezing season. I cordially invite you to tone your statements about the climatology into this direction rather than saying, well, all data sets look just fine with some minor differences.

Based on the earlier comments about the NESOSIM climatology, and the inclusion of our new analysis, we have decided to drop this from the manuscript so this is now redundant.

L403/404: "In general ..." --> I suggest to refer to a table or figure in your manuscript which supports this very general comment.

Yes this was a bit loosely framed. We have now added some extra lines in the plots to show rel002 thickness and have improved the wording to make it clear the thickness distribution difference is greater in the rel002 to rel003 and onwards runs compared to the use of different NESOSIM versions. We have not dropped the climatology from this discussion in part to increase focus on this point.

L468-470: How do the W99 snow densities compare to your densities? Can you identify hotspots in space and/or time where the W99 snow density would be considerably off compared to your results?

A NESOSIM vs mW99 snow density analysis was included in the original Petty et al., (2020) paper (Figures S2-S4). We have now included a reference to this in the revised paper.

"Note that differences between NESOSIM and mW99 snow density are given in P2020 (Supplemental Figures S2-S4). The difference between NESOSIM v1.0 and v1.1 snow density is minimal (as this was not the focus of the v1.1. upgrades) and is expected to have a negligible impact on our thickness results, so we opt against an additional density comparison here."

L478-481: "For example: thinner ... mean thickness." --> consider revisiting and perhaps correcting these statements in light of so far failing to adequately discriminate between total (sea ice plus snow) freeboard and sea ice freeboard in your manuscript.

See above, we were following the ATL10/NSIDC convention here of 'ATL10 sea ice freeboard' but have made various edits based on your suggestions.

L486-510: I am sorry, but I do not see the added value of anomalies computed from three winters of data. To my opinion this Figure 12 and this paragraph can be deleted without substantially changing the relevance of the manuscript.

If you decide to keep both, then I recommend to condense the text considerably and focus on those highlights that seem most obvious to be related with each other. Please keep in mind that a positive snow thickness anomaly in April does not need to coincide with a negative sea ice thickness anomaly - simply because the snow thickness during earlier in the winter determine how much the ice had the chance to grow thermodynamically. There is a temporal dimension involved which cannot be interpreted from the maps shown. Also consider to mention the retrieval noise of the parameters presented to foster readers to disentangle what is noise from what is a real signal.

We would like to keep these maps as we believe they are quite informative. We agree that much of our discussion can be condensed as we have now done, focussing on the main features, and we have also tried to make it clear that when we talk about the impact of snow here we are really referring to the impact on our thickness retrievals, you are right that snow also provides a time-varying impact on the thickness evolution we do not describe here! By reducing this we have also added more weight to the potential uncertainties with NESOSIM in regions like the Barents Sea.

New text: "The most notable feature of these maps are the positive freeboard and thickness anomalies in 2018-2019 and negative freeboard and thickness anomalies in 2020-2021 north of Greenland and the Canadian Arctic Archipelago (CAA), the region of the Arctic where we generally expect to observe the thickest freeboard, snow depth and thickness. On more regional-scales, there are noteworthy examples of the impact of the time-varying NESOSIM snow depths on our thickness retrievals, e.g., the positive snow depth anomalies in the Laptev Sea and the Central Arctic in 2018-2019 modulate the impact of the positive ATL10 freeboard anomalies in 2020-2021 modulate the negative ATL10 freeboard anomalies."

L514-516: "In general ... thicker" --> Does this intercomparison result fit to other results where, e.g. PIOMAS data were compared to in-situ, sub-marine and ICESat data? In my mind there is this result from somewhere in the published literature that PIOMAS over-estimates thinner ice and under-estimates thicker ice - a result that is not that well confirmed by your results - particularly not for the second winter period.

Similar results are shown in Schweiger et al., (2011) in comparisons with the original ICESat mission – e.g., PIOMAS thinner than ICESat in the thickest ice regime and thicker in the thinner ice regime. Similar results were also shown in Wang et al., (2016). A similar, albeit weaker – regional difference pattern has been noted in comparisons with CryoSat-2 (Wang et al., 2016, Petty et al., 2018). We now include a reference to this in the revised manuscript. This section has also undergone changes based on our introduction of new CryoSat-2 comparisons.

L545-551: "These ice type .. this limited record" --> What I am clearly missing here in the discussion is the role of different snow thickness conditions. And this discussion could be also linked better with the 2m-air temperature and longwave radiation data from ERA5.

- What is interesting to see, for instance, is that the sea ice thickness increase between November and February is larger in 2018/19 than in 2019/20 despite a) the sea ice itself being thicker and b) the snow thickness being larger by about 5 cm. Wouldnt' one expect that thicker ice with a thicker snow load grows less thermodynamically than comparably thin ice with a thinner snow load - provided the oceanic and atmospheric forcings are roughly the same?

We think this comment is in relation to the MYI graph, as that is the only one that shows notable ice thickness differences. We don't see that thickness increase as being notably different to the other years considering the uncertainties involved so have refrained from discussion that idea in the paper. However there is perhaps some suggestion of faster ice thickness increase through the middle of winter which is perhaps counter-intuitive given the fact the ice is thicker. It is worth noting however that this is showing ice thickness change, not growth, so dynamics (convergence into the region) may be a likely cause. Our drift maps do not show anything clearly obvious in this regard but this is something an interested user could explore in the Jupyter book! - Another observations is that from November through January the freeboard values are the same for 2019/20 and 2020/21. At the same time snow thickness is about 5 cm larger for 2020/21. This means the sea-ice freeboard which is your total (sea ice plus snow) freeboard minus snow thickness is actually smaller by 5 cm for 2020/21 than 2019/20 which would point to about 40 cm thinner sea ice during these months in 2020/21 than 2019/20 and is actually confirmed by the respective panel. So here your observations are consistent in themselves. The fact that over these 3 months there is 20 cm more ice growth in 2020/21 compared to 2019/20 despite a more or less constant snow load could be discussed more specifically in light of the balance between thicker snow isolating better while thinner sea ice isolates less and the ancillary data used (drift, ERA5).

We have added an extra note on this in the paper as it is a good highlight of the impact of dynamic snow loading:

The MYI results also highlight the important role of dynamic snow loading in constraining regional/monthly thickness variability, e.g., the November-December 2019-2020 and 2020-2021 freeboards are near identical, but NESOSIM indicates 2020-2021 snow depths are  $\sim$  5 cm thicker, resulting in a  $\sim$ 30 cm thicker ice estimate.

- I guess my recommendation is to discuss specific observations in your data time series in a comprehensive way, taking ERA5 and drift information of Fig. 11 into account instead of kind of listing what is shown in the respective panels without connecting the information well to your observations.

We have reworded some of this analysis and also moved this figure to the Supplement, in-part to keep the focus on the thickness data and the extra comparisons we have introduced, and also because to make it clear this is not an exhaustive analysis, but a simple assessment of the link between thickness variability and core atmospheric conditions.

- Did you check, by the way, how ERA5 treats sea ice and its snow cover? How well is the snow thickness on top of sea ice resolved and is the sea ice allowed to grow and melt? Has it a variable thickness that would support discussions about feedback between 2m-air temperatures and ice and snow thickness?

ERA5, as in other reanalyses, has a simple fixed sea ice representation (1.5 m thickness, no snow but fixed albedos). This has impacts on possible warm biases which have been noted in previous studies. We've now made a reference to this in the data and analysis section of the revised manuscript.

L562-564: "For example ... " --> As always the interpretation of such maps offers potential for subjectivity and might depend stongly on the observer. I for my part rather see that the 2019/20 winter is different in terms of the strength and extent of the Beaufort Gyre - including the drift speed of its southern limb - as compared to the other two winters considered. Since the anomaly maps of these two other winters do not provide consistent information, i.e. exhibit different spatial patterns and signs of the anomalies, I am not convinced that the possible relationship between these anomalies and the ice drift pattern should be discussed the way you did. I note in this context, that the anomalies you are referring to here are mostly below 0.5 m in magnitude. How large an anomaly needs to be to be larger than the retrieval noise?

We have tried to not be overly firm in our conclusions we have drawn from analyses such as this. We are not sure there is much we can really change here as we think it best for the reader to further interpret these results beyond the analysis we have presented. L615-617: This statement is not sufficiently well supported by the results given in the manuscript.

We have changed this from accurate to 'important role of dynamic snow loading' instead.

L619/620: "although ... scales." --> Also this part is not sufficiently backed up by the results presented in your manuscript.

We think our analysis did show the differences in seasonal cycles and we noted the regional differences (that are available in the Jupyter Book and have been highlighted in previous studies we now cite based on an earlier comment).

L638-644: I am wondering whether you perhaps could be a bit more specific here because there are snow depth observations from several sovjet drift stations and in addition the N-ICE2015 and the MOSAiC campaigns provided quite a lot of useful snow thickness data to be used for a better calibration and/or evaluation of NESOSIM.

Good idea, we have added the following: "The comprehensive in-situ snow observations collected from recent campaigns, including the Multidisciplinary drifting Observatory for the Study of Arctic Climate (MOSAiC) expedition (Wagner et al., 2022) can hopefully aid with the continued refinement of these new snow reconstructions and redistribution methods."

L556-674: I am not sure this "sea ice thickness reconciliation" paragraph should be kept as is. My impression is that it could be potentially misleading. What I am missing is i) a more clearly formulated statement that ICESat-2 could perhaps be the benchmark sensor for sea-ice thickness retrieval rather than CryoSat-2 - simply because of the more well defined main reflection horizon and the smaller number of snow processes and properties influencing already the freeboard retrieval. ii) What could also be more in the focus of future developments is a better handling of the different spatio-temporal scales involved - both in the retrieval process but also in the evaluation of the products. All parameters, freeboard, snow and ice thickness have their specific distributions. Two issues that I do not find solved satisfactorily is the error propagation of the downscaling approach from 100 km to 30 m and the propagation of the influence of substantial difference in the acquisition time of ICESat-2 data within one grid cell. These (and others) would be enough material for further improvement and I currently do not see the need to write that much about the potential of radar altimetry. But this is clearly my personal view based on this manuscript and on the results presented in similar papers.

We included more discussion of CS-2 issues in part due to previous reviewer comments and also the increased use of CS-2 comparisons in this revised manuscript. We think these are good points to highlight.

We appreciate the ideas you include here and have attempted to include some of these in a revised section, e.g., a comment on ATL10 total freeboard providing a limit on snow loading, and the need to better understand uncertainties and error propagation.

Figure 1: I don't find this figure particularly well developed. Neither is clear that sea ice comprises level and ridged parts nor is clear that a radar measures sea ice freeboard while a laser measures total (sea ice plus snow) freeboard. The fact that the laser signal might also penetrate into the snow a bit is neglected. The role of the "internal ice stresses" is not clear, neither is mentioning of keel depth for satellite altimetry of sea ice. Not represented well is that radar waves may in fact be reflected at the snow-ice interface or even from within the sea ice but that they may also not reach to that interface at all. While

the figure is for winter, which is good, the caption refers to the key challenges and one of these definitely is to measure freeboard during summer in the presence of melt ponds. So there is in fact a lot more one could and potentially should include into this schematic figure.

It is obviously challenging to include all those issues and processes in one simple diagram. We've tried to identify the main issues in our view – the laser (total) and radar (ice) freeboard profiling, clouds scattering laser, variable topography, but we appreciate the feedback in trying to improve this.

We have thus adapted the figure to include extra issues affecting sea ice altimetry. We have also added winter to the caption as it was always our intention to focus on winter issues here. Please see updated Figure 1.

Figure 2: Particularly in the bottommost row the white circular area is not a well defined circle. What is the reason for that? Is there a higher probability for data drop-outs near the pole? The main aim of the paper is to show the differences between rel002 used Petty et al. 2020 and rel005 used in this paper. I am wondering what can learn in terms of science from the many other panels and whether it wouldn't be sufficient and more straightforward to concentrate on the differences between rel002 and rel005 only.

The uneven hole is just a function of the binning procedure. We have since simplified this figure to show just a single row of rel002 to rel005 coverage then just the rel002 to rel005 difference. We have included the original figure (which includes all rel002 to rel005 differences) in the SI.

Figure 3: What about the median biases for both cases? I note that (b) shows more cases than (a). Why?

Mean bias is a common diagnostic for comparisons like this so we prefer to stick with that for clarity. The SD captures deviations beyond that. The reason b showed more grid-cells in the comparison is because of slight differences in the number of valid grid-cells from the different configuration. As 2 is negligible in terms of its impact on these statistics, we have dropped N from the panels.

- Personally, I would make the lengths of x- and y-axes the same when the data range is the same - here 80 cm for both quantities shown.

We did not have time to re-do this figure but the aspect ratio seems close enough to 1:1.

Figure 4 c) What is the reason for the violins for Oct. through Jan. to look cut off at the side facing higher snow thickness values?

This is simply a product of the KDE fitting procedure (the violin plot cuts off outside of the raw data input), but is still able to highlight the skewness of the distributions.

Figure 5: This map contains many regions that are not used in the paper. It might make sense to color all regions not used in the same color as open water, only explain the acronyms / names of the regions actually used, and add that this map is "adopted" from a region mask provided by ...

Agreed, we have changed this figure accordingly and decided to keep the colors but only annotate the regions used in our study domain.

Figure 6: The x-axis title needs to be changed into "total freeboard". Please provide the meaning of the dashed lines in the caption.

**Dashed lines are the means and we have now included that in the caption. Thanks!**

- I strongly recommend to be consistent in the notation between the figure and the caption. You write r002, r003 and so forth in the figure but write "rel002" and "rel005" in the caption text; this is inconsistent. Similarly, in the figure you write "bnum1", "bnum3" and so on, in the caption text you write "beam #1, #3" and so on. This is inconsistent as well.

**Good point, we have rectified this in the revised version.**

Figure 9: I note that the sea ice thickess uncertainty seems not to be influenced by the substantial differences in the "mean day of the month" varying over a very short distance the ICESat-2 data. Hence in one 25 km grid cell ICESat-2 might come (on average) from the first few days of a month while in the adjacent grid cell this data might stem from the end of the month. I would expect that thermodynamic sea ice growth and dynamic processes can add substantially to the uncertainty - most likely especially during months October through January. Did you check this?

This is a good point, and is something we have considered, although it is hard to exactly quantify this. It is our belief that with more data we can start to understand expected monthly changes in thickness and quantify this sampling error, which we hope to include in a future data release. *"We hope to incorporate a more comprehensive and accurate accounting of the various error contributions where possible, e.g., accounting for the clear sampling/representation error associated*

with grid-cell data differences, while also exploring more sophisticated uncertainty quantification methods (e.g. Monte Carlo approaches)".

Figure 14: I guess it would make sense to not show values for September but rather begin with October as this is when the freezing season commences. There is room for one more panel in which you could put the fraction of the FYI relative to the total sea ice area of your Inner Arctic region. The same comment applies to Figure 15, but here for the MYI fraction of course.

This is a good point, we have dropped September from the FYI figure and made a note on this in the caption and manuscript. We don't feel the need for another panel on FYI vs total area as interested readers can make this out from the MYI fraction in Figure 10.

**Pure typos / editoral comments:**

A general editoral comment upfront: Since you advertize at the end that basically all your results and the processing is available as jupyter notebook you might want to scan you paper and reduce the number of repeated mentioning of this. I think you overdo it a little bit.

L28/29: "Mean first-year ... negligible" --> possibly you refer to changes or differences?

Added differences, thanks.

L139: "2we" --> "2, we"

**Changed, thanks.**

L218: "consensus" --> In the modeling world one would speak of an ensemble mean - or in your case apparently an ensemble median. I would find this a better expression because "consensus" is something that usually involves some discussion, some weighing perhaps, some additional constraints. What you

do, however, is taking mean and median, i.e. use a statistical tool. Hence, I would recommend to change the wording accordingly - here and later in the text.

Agreed, changed this language to refer mainly to 'median' gridded OIB. data.

L231: "halving the blowing snow open water coefficient" --> would you mind to show the respective equation in this paper as well so that everybody immediately understand the effect of what you stated here?

Yes we have now added a new Eq. 1 to the revised manuscript.

L247-249: You are providing enough information about this region map in the context of Figure 5 I guess and can delete the sentence "Out Inner ... Figure 5" and instead refer to this figure at the end of the previous sentence.

**Agreed, removed.**

L304-306: "Our monthly ... ATL10 data" --> namely what ancillary data?

The ancillary data is what follows in this sentence, the mean day and number of segments.

L309: "on to" --> "onto"

We do not think this correct so have kept the language as is.

L311-317: This reads a lot like again being in a technical report. Have you thought about publishing this manuscript in the journal Earth System Science Data? Would that be a more adequate journal for your manuscript being comparably rich in processing and product details, bug fixes, novel masks, and so forth? I slowly begin to second that other reviewer who had difficulties to see the merit of this manuscript in The Cryosphere.

See initial discussion and new analysis included!

L374-376: Since you do not consider June I suggest to remove June and the respective value from this sentence.

Agreed, removed.

L407: Here you write rel003, in L413++ you write rel005. What is correct?

It should be rel005, this was changed in the revisions but not updated here. Corrected now, thanks for spotting that.

L413-424: I am wondering whether you need to list all ranges for all parameters shown in Figure 8. Do you see a chance to simply let speak Figure 8 for itself and perhaps only put the highlights here - perhaps along the lines that correlation coefficients in general increased by around 0.1, that biases change (not necessarily reduce) by about 30 cm and that standard deviations reduce by, on average, 0.2 m. This might make this part of the paper easier to read.

Agreed, we have condensed this dramatically as we have now included the new CryoSat-2 thickness comparisons. See discussion at the start.

L417: "58" --> "0.58" Thanks, changed.

L434: "i-j" --> "i-k"

Thanks, changed.

L450/451: "The data within this domain ... Figure 10e)" --> You need to put labels a) to f) to the panels in Figure 10 if you want to refer to them.

Thanks, added.

- I cannot see any sea-ice concentrations within the range given here (40-60%) in your Figure 10 e; something needs to be corrected here.

Thanks, this was an error that has been corrected in the revised paper.

L533: "snow depth in November ... winters" --> you could denote in addition, that in Feb to April, however, differences in the snow thickness on the sea ice are between 2 cm and 4 cm.

Agreed, we've added this extra detail now.

L611: I suggest to use "significant" only in the context of statistical tests which I could not see in your manuscript. Hence "considerable", "notable", or "substantial" might be a better wording here.

Agreed, we've removed as much use of significance here and elsewhere and replaced with phrase slike notable, or large.

L654/655: "The extension ... estimates" --> While this is true it would require first and foremost that we have (more) accurate freeboard estimates during summer from ICESat-2?

Thanks, we've added this clarification and a comment on the recent summer cal/val campaign. "together with improved understanding of the performance of ICESat-2 over the complex summer melt surface for summer freeboard determination (Tilling et al., 2020). A recently completed 2022 ICESat-2 summer airborne cal/val campaign should also provide important insights towards this goal."

L920: "now depths" --> "snow depths"

Thanks, changed.

L958/959: Perhaps better (and more correct): "The background dark grey shading in panels a to k is the CDR sea ice concentration shown in panel l."

Thanks, changed.

References

Glissenaar, I. A., J. C. Landy, A. A. Petty, N. T. Kurtz, and J. C. Stroeve (2021), Impacts of snow data and processing methods on the interpretation of long-term changes in Baffin Bay sea ice thickness, The Cryosphere, 15, 4909–4927, doi:10.5194/tc-15-4909-2021.

Landy, J. C., Dawson, G. J., Tsamados, M., Bushuk, M., Stroeve, J. C., Howell, S. E. L., Krumpen, T., Babb, D. G., Komarov, A. S., Heorton, H. D. B. S., Belter, H. J., and Aksenov, Y.: A year-round satellite sea-ice thickness record from CryoSat-2, Nature, 609, 517–522, https://doi.org/10.1038/s41586-022-05058-5, 2022.

Ricker, R., Hendricks, S., Kaleschke, L., Tian-Kunze, X., King, J., and Haas, C.: A weekly Arctic sea-ice thickness data record from merged CryoSat-2 and SMOS satellite data, The Cryosphere, 11, 1607–1623, https://doi.org/10.5194/tc-11-1607-2017, 2017.

Schweiger, A., Lindsay, R., Zhang, J., Steele, M., Stern, H., and Kwok, R. (2011), Uncertainty in modeled Arctic sea ice volume, *J. Geophys. Res.*, 116, C00D06, doi:10.1029/2011JC007084.

Wang, X.; Key, J.; Kwok, R.; Zhang, J. Comparison of Arctic Sea Ice Thickness from Satellites, Aircraft, and PIOMAS Data. Remote Sens. 2016, 8, 713. https://doi.org/10.3390/rs8090713

---

## Author Response (AR3)

We sincerely thank the editor and reviews for the time and effort taken to oversee our manuscript. The paper has improved significantly and we appreciate the help to get us to this point.

In this final submission we have included just a track changes of the small edits made based on the final review comments.

Alek Petty